# Understanding the Robustness of Randomized Feature Defense Against Query-Based Adversarial Attacks

**Quang H. Nguyen**[1]**, Yingjie Lao**[2]**, Tung Pham**[3]**, Kok-Seng Wong**[1]**, Khoa D. Doan**[1]
[1]College of Engineering and Computer Science, VinUniversity, Vietnam
[2]Tufts University
[3]VinAI Research
`quang.nh@vinuni.edu.vn, yingjie.lao@tufts.edu, v.tungph4@vinai.io`
`wong.ks@vinuni.edu.vn, khoa.dd@vinuni.edu.vn`

## Abstract

Recent works have shown that deep neural networks are vulnerable to adversarial examples that find samples close to the original image but can make the model misclassify. Even with access only to the model's output, an attacker can employ black-box attacks to generate such adversarial examples. In this work, we propose a simple and lightweight defense against black-box attacks by adding random noise to hidden features at intermediate layers of the model at inference time. Our theoretical analysis confirms that this method effectively enhances the model's resilience against both score-based and decision-based black-box attacks. Importantly, our defense does not necessitate adversarial training and has minimal impact on accuracy, rendering it applicable to any pre-trained model. Our analysis also reveals the significance of selectively adding noise to different parts of the model based on the gradient of the adversarial objective function, which can be varied during the attack. We demonstrate the robustness of our defense against multiple black-box attacks through extensive empirical experiments involving diverse models with various architectures. Code is available at `https://github.com/mail-research/randomized_defenses`

## 1 Introduction

Modern deep neural networks have demonstrated remarkable performance in various complex tasks, including image classification and face recognition, among others. However, prior works have pointed out that deep learning models are sensitive to small changes in the input and can be fooled by carefully chosen and imperceptible perturbations Szegedy et al. (2014); Goodfellow et al. (2015); Papernot et al. (2016b); Madry et al. (2018). These adversarial attacks can be generally classified into white-box and black-box attacks. In a white-box setting, strong attacks such as Projected Gradient Descent (PGD) Madry et al. (2018) can generate effective adversarial examples by levering the information inside the model. However, in practical scenarios such as machine learning as a service (MLaas), the well-trained models and the training datasets are often inaccessible to the users, especially in the era of large models. Hence, query-based black-box attacks become the primary threats in most real-world applications, where the adversary is assumed to have no knowledge of the model architecture and parameters.

This paper proposes a lightweight, plug-and-play defensive method that can significantly decrease the success rate of query-based black-box attacks, including both score-based and decision-based attacks Ilyas et al. (2018; 2019); Andriushchenko et al. (2020); Guo et al. (2019); Al-Dujaili & O'Reilly (2020); Liu et al. (2019); Chen & Gu (2020); Chen et al. (2020b); Rahmati et al. (2020). Adversarial examples generated through query-based attacks involve iterative procedures that rely on either local search techniques involving small incremental modifications to the input or optimization methods leveraging estimated gradients of the adversary's loss concerning the input. However, the process of requesting numerous queries is time-consuming and sometimes may raise suspicions with the presence of multiple similar queries. Hence, the objective of defense is to perplex the adver-

sary when attempting to generate adversarial examples. Our proposed method accomplishes this by introducing noise into the feature space. While Qin et al. (2021) study the robustness of randomized input, this paper provides both theoretical analysis and empirical evidence to demonstrate improved robustness of randomized features. Our analysis also highlights the importance of strategically introducing noise to specific components of the model based on the gradient of the adversarial objective function, which can be dynamically adjusted throughout the attack process.

Our contributions can be summarized as follows:

- We investigate the impact of randomized perturbations in the feature space and its connection to the robustness of the model to black-box attacks.

- We design a simple yet effective and lightweight defense strategy that hampers the attacker's ability to approximate the direction toward adversarial samples. As a result, the success rate of the attacks is significantly reduced.

- We extensively evaluate our approach through experiments on both score-based and decision-based attacks. The results validate our analysis and demonstrate that our method enhances the robustness of the randomized model against query-based attacks.

## 2 RELATED WORKS

### 2.1 ADVERSARIAL ATTACKS

Extensive research has been conducted on white-box attacks, focusing on the generation of adversarial examples when the attacker possesses complete access to the target model. Over the years, various notable methods have emerged as representative approaches in this field, including fast gradient sign method (FGSM) Goodfellow et al. (2015), Jacobian-based saliency Map Attack (JSMA) Papernot et al. (2016a), and PGD Madry et al. (2018).

In contrast to white-box attacks, the black-box scenario assumes that the attacker lacks access to the target model, making it a more challenging situation. However, this is also a more realistic setting in real-world applications where the adversary would not have access to the model parameters. One approach in black-box attacks involves utilizing white-box techniques on substitute models to create adversarial examples, which can subsequently be applied to black-box target models Papernot et al. (2017). However, the effectiveness of transfer-based attacks can vary significantly due to several practical factors, such as the initial training conditions, model hyperparameters, and constraints involved in generating adversarial samples Chen et al. (2017). This paper focuses on the defense against query-based attacks instead.

### 2.2 QUERY-BASED BLACK-BOX ATTACKS

Query-based attacks can be largely divided into score-based attacks and decision-based attacks, based on the accessible model output information. Score-based attacks leverage the output probability or logit of the targeted model, allowing the attacker to manipulate the scores associated with different classes. On the other hand, decision-based queries provide the attacker with hard labels, restricting the access to only the final predictions without any probability or confidence values.

We list the query-based attacks used in this paper below:

**Natural Evolutionary Strategies (NES) Ilyas et al. (2018)** is one of the first query-based attacks that use natural evolutional strategies to estimate the gradient of the model with respect to an image $x$. By exploring the queries surrounding $x$, NES effectively gauges the model's gradient, enabling it to probe and gain insights into the model's behavior.

**SignHunt Al-Dujaili & O'Reilly (2020)** is another score-based attack, which flips the sign of the perturbation based on the sign of the estimated gradient to improve the query efficiency.

**Square attack Andriushchenko et al. (2020)** is a type of score-based attack that differs from gradient approximation techniques. Instead, it employs random search to update square-shaped regions located at random positions within the images. This approach avoids relying on gradient information and introduces a localized square modification to the image.

**RayS Chen & Gu (2020)** is a decision-based attack that solves a discrete problem to find the direction with the smallest distance to the decision boundary while using a fast check step to avoid unnecessary searches.

**SignFlip Chen et al. (2020b)** is an $\ell^\infty$ decision based attack that alternately projects the perturbation to a smaller $\ell^\infty$ ball and flips the sign of some randomly selected entries in the perturbation.

## 2.3 DEFENSIVE METHODS AGAINST QUERY-BASED ATTACKS

In the recent literature, several defensive solutions have been proposed to counter adversarial examples. One such solution involves the detection of malicious queries by comparing them with previously observed normal queries Chen et al. (2020a); Li et al. (2022); Pang et al. (2020). This approach aims to identify anomalous patterns in queries and flag them as potential adversarial examples. Additionally, adversarial training has also been utilized to enhance the model's robustness Cohen et al. (2019); Wang et al. (2020); Sinha et al. (2017); Zhang et al. (2020). Adversarial training involves training the model on both regular and adversarial examples to improve its ability to withstand adversarial attacks. However, it is computationally expensive, especially when dealing with large and complex datasets. In some cases, adversarial training may also inadvertently harm the model's overall performance.

In contrast, this paper focuses on approaches that involve incorporating noise or randomness into the model, thereby providing the adversary with distorted information. The underlying intuition behind these defense mechanisms is to deceive the attacker by introducing perturbations in the model's prediction process. By altering certain signals, the defenses aim to mislead the attacker and divert them from their intended direction. To achieve this, various techniques are employed to modify the input data or manipulate the model's internal workings. For instance, some defenses may introduce random noise or distortion to the input samples Liu et al. (2017); He et al. (2019); Salman et al. (2019), making them less susceptible to adversarial perturbations. This noise acts as a smokescreen, confusing the attacker and making it harder for them to generate effective adversarial examples.

We list the defensive methods evaluated in this paper below:

**Random Noise Defense (RND) Qin et al. (2021)** is a lightweight defense that adds Gaussian noise to the input for each query. This work also theoretically shows RND's effectiveness against query-based attacks. Byun et al. (2021) proposes Small Noise Defense (SND), whose randomization method is identical to RND, thus reporting performance valuation for RND covers both of these works.

**Adversarial Attack on Attackers (AAA) Chen et al. (2022)** directly optimizes the model's logits to confound the attacker towards incorrect attack directions.

## 3 METHOD

### 3.1 PROBLEM FORMULATIONS

**Adversarial attack.** Let $f : \mathbb{R}^d \to \mathbb{R}^K$ be the victim model, where $d$ is the input dimension, $K$ is the number of classes, $f_k(x)$ is the predicted score of class $k$ for input $x$. Given an input example $(x, y)$, the goal of adversarial attack is to find a sample $x'$ such that

$$\arg \max_k f(x') \neq y, \quad \text{s.t} \quad d(x, x') \leq \epsilon, \tag{1}$$

where $d(x, x')$ is distance between samples $x$ and $x'$. In practice, the distance can be the $\ell^2-$norm, $\|x - x'\|_2$, or the $\ell^\infty-$norm, $\|x - x'\|_\infty$.

This adversarial task can be framed as a constrained optimization problem. More particularly, the attacker tries to solve the following objective

$$\min_{x'} \mathcal{L}(f(x'), y), \quad \text{s.t} \quad d(x, x') \leq \epsilon, \tag{2}$$

where $\mathcal{L}(.,.)$ is a loss function designed by the attacker. In practice, a common loss function $\mathcal{L}$ is the max-margin loss, as follows:

$$\mathcal{L}(f(x), y) = f_y(x) - \max_{i \neq y} f_i(x). \tag{3}$$

**Score-based attack.** For the query-based attack, an attacker can only access the input and output of the model; thus, the attacker cannot compute the gradient of the objective function with respect to the input $x$. However, the attacker can approximate the gradient using the finite difference method:

$$\hat{\nabla}\mathcal{L} = \sum_u \frac{\mathcal{L}(f(x+\eta u), y) - \mathcal{L}(f(x), y)}{\eta} u, \quad \text{where } u \sim \mathcal{N}(0, \mu I). \tag{4}$$

Another approach to minimize the objective function is via random search. Specifically, the attacker proposes an update $u$ and computes the value of $\mathcal{L}$ of this update to determine if $u$ can help improve the value of the objective function. Formally, the proposed $u$ is selected if $\mathcal{L}(f(x+u), y) - \mathcal{L}(f(x), y) < 0$, otherwise it is rejected.

**Decision-based attack.** In contrast to score-based attacks, hard-label attacks find the direction that has the shortest distance to the decision boundary. The objective function of an untargeted hard-label attack can be formulated as follows:

$$\min_d g(d)$$
$$\text{where} \quad g(d) = \min \left\{ r : \arg\max_k f(x + rd/\|d\|_2) \neq y \right\}. \tag{5}$$

This objective function can be minimized using binary search, in which the attacker queries the model to find the distance $r$ for a particular direction $d$. To improve the querying efficiency, binary search can be combined with fine-grained search, in which the radius is iteratively increased until the attacker finds an interval that contains $g(d)$. Hence, the gradient of $g(d)$ can also be approximated by the finite difference method

$$\hat{\nabla} g(d) = \sum_u \frac{g(d+\eta u) - g(d)}{\eta} u. \tag{6}$$

Similar to the case of score-based attacks, the attacker can also search for the optimal direction. Given the current best distance $r_{\text{opt}}$, a proposed direction $d$ is eliminated if it cannot flip the prediction using the current best distance $r_{\text{opt}}$; otherwise the binary search is used to compute $g(d)$, which is the new best distance.

**Randomized model.** In this work, we consider a randomized model $f_{\text{rand}} : \mathbb{R}^d \to \mathcal{P}(\mathbb{R}^K)$ that maps a sample $x \in \mathbb{R}^d$ to a probability distribution on $\mathbb{R}^K$. Given an input $x$ and an attack query, the corresponding output is a vector drawn from $f_{\text{rand}}(x)$. We assume that the randomized model $f_{\text{rand}}$ is 'nice'; that is, the mean and variance of $f_{\text{rand}}(x)$ exist for every $x$.

Finally, we define adversarial samples for a randomized model. Since the model has stochasticity, the prediction returned by the model of a sample $x$ can be inconsistent at different queries; i.e., the same sample can be correctly predicted at one application of $f_{\text{rand}}$ and be incorrectly predicted later in another application of $f_{\text{rand}}$. For this reason, adversarial attacks are successful if the obtained adversarial example can fool the randomized model in the majority of its applications on the example.

**Definition 1** (Attack Success on Randomized Model). *Given a data point $x$ with label $y$ and a positive real number $\epsilon$, a point $x'$ is called adversarial samples in a closed ball of radius $\epsilon$ around $x$ with respect to the model $f_{\text{rand}}$ if $\|x' - x\|_p < \epsilon$ and*

$$\arg\max \mathbb{E}[f_{\text{rand}}(x')] \neq y.$$

## 3.2 RANDOMIZED FEATURE DEFENSE

Our method is based on the assumption that the attacker relies on the model's output to find the update vector toward an adversarial example. Consequently, if the attacker receives unreliable feedback from the model, it will be more challenging for the attacker to infer good search directions toward the adversarial sample.

In contrast to the previous inference-time randomization approaches, we introduce stochasticity to the model by perturbing the hidden features of the model. Formally, let $h_l$ be the $l-$th layer of the model, we sample an independent noise vector $\delta$ and forward $h_l(x) + \delta$ to the next layer. For simplicity, $\delta$ is sampled from Gaussian distribution $\mathcal{N}(0, \Sigma)$, where $\Sigma$ is a diagonal matrix, or $\mathcal{N}(0, \nu I), \nu \in \mathbb{R}$. The detailed algorithm is presented in Algorithm 1.

Let $f_{\mathrm{rand}}$ be the proposed randomized model corresponding to the original $f$. When the variance of injected noise is small, we can assume that small noise diffuses but does not shift the prediction. Particularly, we can make the following assumption.

**Assumption 1.** *Mean of the randomized model $f_{\mathrm{rand}}$ with input $x$ is exactly the prediction of the original model for $x$*

$$\mathbb{E}[f_{\mathrm{rand}}(x)] = f(x).$$

Given Assumption 1, by Definition 1, adversarial samples of the original model are adversarial samples of the randomized model. Therefore, the direction that the attacker seeks is also that of the original model. Recall that the attacker finds this direction by either finite difference or random search.

---

**Algorithm 1** Randomized Feature Defense

**Input:** a model $f$, input data $x$,
    noise statistics $\Sigma$, a set of perturbed layers
    $H = \{h_{l_0}, h_{l_1}, \ldots, h_{l_n}\}$
**Output:** logit vector $l$
    $z_0 \leftarrow x$
    **for** layer $h_i$ in the model **do**
        **if** $h_i \in H$ **then**
            $\delta \sim \mathcal{N}(0, \Sigma)$
            $z_i \leftarrow h_i(z_{i-1}) + \delta$
        **end if**
    **end for**
    $l \leftarrow z_n$

---

In our method, when the model is injected with an independent noise, the value of objective $\mathcal{L}$ is affected. If $\mathcal{L}(f_{\mathrm{rand}}(x + \eta u), y) - \mathcal{L}(f_{\mathrm{rand}}(x), y)$ oscillates among applications of $f_{\mathrm{rand}}$, the attacker is likely misled and selects a wrong direction. For random-search attacks, when the sign of $\mathcal{L}(f_{\mathrm{rand}}(x + \eta u), y) - \mathcal{L}(f_{\mathrm{rand}}(x), y)$ and the sign of $\mathcal{L}(f(x + \eta u), y) - \mathcal{L}(f(x), y)$ are different, the attacker chooses the opposite action to the optimal one. In other words, the attacker can either accept a bad update or reject a good one in a random search.

### 3.3 Robustness to Score-based Attacks

In this section, we present the theoretical analysis of the proposed defense against score-based attacks.

**Theorem 1.** *Assuming the proposed random vector $u$ is sampled from a Gaussian $\mathcal{N}(0, \mu I)$, the model is decomposed into $f = g \circ h$, and the defense adds a random noise $\delta \sim \mathcal{N}(0, \nu I)$ to the output of $h$. At input $x$, the probability that the attacker chooses an opposite action positively correlates with*

$$\arctan\left(-\left(\frac{2\nu}{\mu}\frac{\|\nabla_{h(x)}(\mathcal{L} \circ g)\|_2^2}{\|\nabla_x(\mathcal{L} \circ f)\|_2^2}\right)^{-0.5}\right).$$

This theorem states that the robustness of the randomized model is controlled by both (i) the ratio between the defense and attack noises and (ii) the ratio of the norm of the gradient with respect to the feature $h(x)$ and the norm of the gradient with respect to the input $x$. Since $\arctan$ is monotonically increasing, the model becomes more robust if the ratio $\frac{2\nu}{\mu}\frac{\|\nabla_{h(x)}(\mathcal{L} \circ g)\|_2^2}{\|\nabla_x(\mathcal{L} \circ f)\|_2^2}$ is high. Intuitively, the perturbations added by the attacker and by the defense induce a corresponding noise in the output; if the attack noise is dominated by the defense noise, the attacker cannot perceive how its update affects the model. Note that the $\arctan$ function is bounded, which means at some point the robustness saturates when the ratio increases.

While the first ratio is predetermined before an attack, the second ratio varies during the attack when the input $x$ is sequentially perturbed since it depends on the gradient of the objective function. To understand this behavior of the randomized model during the attack, we perform the following experiment. First, we compute the ratio of the norms of gradients at $h(x)$ and $x$. To simulate an attacker, we perform a single gradient descent step with respect to $\mathcal{L}$. The distributions of the ratios on the raw and perturbed images at different layers are shown in Figure 1. We can observe that these ratios become higher when the data are perturbed toward the adversarial samples. In other words, the randomized model is more robust during the attack. We also illustrate the accuracy under Square attack when adding noise to each layer, verifying our analysis.

### 3.4 Robustness to Decision-based Attacks

In decision-based attacks, the attacker finds the optimal direction $d_{\mathrm{opt}}$ and the corresponding distance $r_{\mathrm{opt}}$ to the decision boundary such that $r_{\mathrm{opt}}$ is minimal. We use the objective function

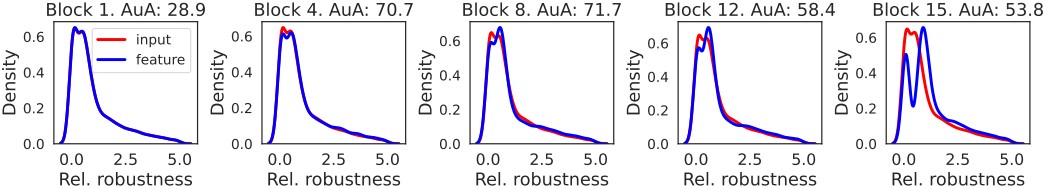

Figure 1: The ratio of the norm of the gradient of $\mathcal{L}$ at selected hidden layers and at input of VGG19 on CIFAR10 before and after perturbed. Full results are provided in the supplementary material.

Figure 2: Distributions of the magnitude of the robustness to query-based attacks computed at input and selected hidden layers of VGG19 on CIFAR10.

$\mathcal{L}(f(x), y)$ to understand how our method affects the decision-based attacks. Indeed, $\mathcal{L}$ measures how close the prediction is to the true label: $\mathcal{L} \leq 0$ if the prediction is incorrect and $\mathcal{L} > 0$ otherwise.

To estimate $g(d)$, the attacker can use binary search. Similar to score-based attacks, when noise is injected into the model, the function $g(d)$ becomes stochastic, which eventually affects the attack. Unfortunately, the distribution of $g(d)$ (under binary search with randomness) does not have an analytical form. Nevertheless, we can still use a similar analysis to the last section to understand the robustness of our method. To avoid performing a binary search on uninformative directions, the attacker relies on best-radius searching. Given the current best distance $r_{\text{opt}}$, for every new direction $d$, the attacker verifies if the distance along $d$ to the boundary is shorter than $r_{\text{opt}}$ by querying $x + r_{\text{opt}} d / \|d\|_2$. When adding noise to features $h(x)$ of $f = g \circ h$ and linearizing the function at the current input $x$, we have

$$\mathcal{L}(f_{\text{rand}}(x + r_{\text{opt}} d / \|d\|_2), y) \approx \mathcal{L}(g(h(x) + r_{\text{opt}} J_h(x) d / \|d\|_2 + \delta), y) \tag{7}$$

$$\approx \mathcal{L}(f(x), y) + r_{\text{opt}} \nabla_x \mathcal{L}(f(x), y) d / \|d\|_2 + \nabla_{h(x)} \mathcal{L}(g(h(x)), y) \delta \tag{8}$$

$$\approx (r_{\text{opt}} - g(d)) \nabla_x \mathcal{L}(f(x), y) d / \|d\|_2 + \nabla_{h(x)} \mathcal{L}(g(h(x)), y) \delta, \tag{9}$$

where $J_h(x)$ is the Jacobian matrix of $h$ evaluated at $x$, since $\mathcal{L}(f(x), y) + g(d) \nabla_x \mathcal{L}(f(x), y) d / \|d\|_2 \approx \mathcal{L}(f(x + g(d) d / \|d\|_2), y) = 0$. If $\delta \sim \mathcal{N}(0, \nu I)$, the variance of $\nabla_{h(x)} \mathcal{L}(g(h(x)), y) \delta$ is $\nu \|\nabla_{h(x)} \mathcal{L}(g(h(x)), y)\|_2^2$. When this value is large, it can dominate the other terms and increase the chance of flipping the sign of the loss function $\mathcal{L}$. In other words, when $\mathcal{L}$ has a high variance, the attacker is more likely to misjudge the direction.

## 3.5 THE EFFECT OF RANDOMIZED FEATURES ON ACCURACY

Let $\mathcal{D}$ be the data distribution, without any attack or defense, the accuracy of the model is

$$\text{Acc}(f) := \mathbb{E}_{(x,y) \sim \mathcal{D}} [\mathbb{1}(f(x) = y)] = \mathbb{E}_{(x,y) \sim \mathcal{D}} [\mathbb{1}(\mathcal{L}(f(x), y) > 0)]. \tag{10}$$

When injecting noise into the model, it becomes a robust, stochastic model $f_{\text{rand}} : \mathbb{R}^d \to \mathcal{P}(\mathbb{R}^K)$.

The clean accuracy of the randomized model is

$$\text{Acc}(f_{\text{rand}}) = \mathbb{E}_{(x,y) \sim \mathcal{D}} \mathbb{E}_{y' \sim f_{\text{rand}}(x)} [\mathbb{1}(y' = y)] = \mathbb{E}_{(x,y) \sim \mathcal{D}} \mathbb{E}_{y' \sim f_{\text{rand}}(x)} [\mathbb{1}(\mathcal{L}(y', y) > 0)]. \tag{11}$$

Adding noise $\delta_2 \sim \mathcal{N}(0, \nu_2 I)$ to the features at layer $h$ of the model $f = g \circ h$ results in:

$$\text{Acc}(f_{\text{rand}}) = \mathbb{E}_{(x,y) \sim \mathcal{D}} \mathbb{E}_{\delta \sim \mathcal{N}(0, \nu_2 I)} [\mathbb{1}(\mathcal{L}(g(h(x) + \delta_2), y) > 0)] \tag{12}$$

$$\approx \mathbb{E}_{(x,y) \sim \mathcal{D}} \mathbb{E}_{\delta_2 \sim \mathcal{N}(0, \nu_2 I)} [\mathbb{1}(\mathcal{L}(f(x), y) + \nabla_{h(x)} (\mathcal{L} \circ g) \delta_2 > 0)] \tag{13}$$

$$= \mathbb{E}_{(x,y) \sim \mathcal{D}} \mathbb{E}_{\delta_2' \sim \mathcal{N}(0, \nu_2)} [\mathbb{1}(\mathcal{L}(f(x), y) / \|\nabla_{h(x)} (\mathcal{L} \circ g)\|_2 + \delta_2' > 0)]. \tag{14}$$

It means that the accuracy of a randomized model depends on the objective function and its gradient, which vary for different data points. These ratios of $\mathcal{L}$ and its gradient computed at the input and hidden layers are different. If $\mathcal{L}$ is small at samples that have a large gradient norm when noise is injected at a layer, these samples will be likely misclassified while the correctly classified samples have a low magnitude of robustness (i.e., $\nu\|\nabla_{h(x)}(\mathcal{L} \circ g)\|_2^2$ is small, as discussed in Theorem 1 and Section 3.4). In contrast, if the gradient norm with respect to the randomized layer is large for samples that have large $\mathcal{L}$, the robustness of the model for the correctly classified samples will be high; thus, adding noise to this layer makes the model more robust against black-box attacks.

We conduct the following experiment to understand how the defense affects the whole dataset. We first compute the ratios of $\mathcal{L}$ and its gradient for all samples and keep the top 99% values. Essentially, the standard deviation of defensive noises that makes the accuracy drop by 1% is proportional to the value at which 1% of the ratios in the dataset are smaller. The product of this value and the norm of gradient represents the robustness of datasets, which are shown in Figure 2. We also illustrate the accuracy under Square attack when adding noise to each layer.

We can observe that the ratio distributions when randomizing the input and the hidden features are similar at the first few layers of the model; however, these ratios at the deeper layers of the model are higher. This means that randomizing the model at these layers makes it more robust than adding noise to the input layer when the defenders desire similar clean accuracy in the randomized models.

## 4 EXPERIMENTS

In this section, we evaluate the empirical performance of the proposed randomized feature defense.

### 4.1 EXPERIMENTAL SETUP

**Datasets**. We perform our experiments on two widely used benchmark datasets in adversarial robustness: CIFAR10 Krizhevsky & Hinton (2009) and ImageNet Russakovsky et al. (2015). We randomly select 1000 images that contain every class from the studied dataset in each experiment.

**Defenses**. In addition to the proposed defense, we also include the related input defenses Qin et al. (2021); Byun et al. (2021) in our evaluation. Note that, the empirical robustness comparison of all adversarial defenses is beyond the scope of the paper since our objective is to theoretically and empirically study the effectiveness of the randomized feature defense. We also evaluate AAA defense Chen et al. (2022) against decision-based attacks and compare them with randomized defenses.

In all experiments, our defense randomizes only the penultimate layers of the base models, since our theoretical observations in Section 3.3 (Figure 1), and empirical results, provided in Table 14 (Supplementary), show that randomizing a deeper layer is consistently more effective.

**Attacks**. For score-based attacks, we consider the gradient-estimation methods, NES Ilyas et al. (2018), and the random-search methods, Square Andriushchenko et al. (2020), SignHunt Al-Dujaili & O'Reilly (2020). For decision-based attacks, we consider RayS Chen & Gu (2020) and Sign-Flip Chen et al. (2020b).

**Models**. We consider 6 victim models on ImageNet, including 2 convolution models that are VGG19 Simonyan & Zisserman (2015) and ResNet50 He et al. (2016), 2 transformer models that are ViT Dosovitskiy et al. (2021) and DeiT Touvron et al. (2021). For the experiments on CIFAR10, we finetuned VGG19, ResNet50, ViT, DeiT with an input size of $224 \times 224$.

**Evaluation protocol**. For a fair comparison, we report each defense's robustness performance results at the corresponding configuration of hyperparameters that achieves a specific drop (i.e., $\approx 1\%$ or $\approx 2\%$) in clean-data accuracy. In practice, a defender always considers the trade-off between robustness and clean-data performance, with a priority on satisfactory clean-data performance; thus, achieving higher robustness but a significant drop in clean-data accuracy is usually not acceptable.

### 4.2 PERFORMANCE AGAINST SCORE-BASED ATTACKS

On ImageNet, we report the accuracy under the attack of 6 models and 3 score-based attacks in Table 1. As we can observe, while the attacks achieve close to 0% failure rate on the base models (i.e., without any defense), both randomized feature and input defenses significantly improve the models' robustness against score-based attacks. Furthermore, for Square attack and SignHunt, which are strong adversarial attack baselines, randomized feature defense consistently achieves better perfor-

mance on all 6 models, which supports our theoretical analysis in Section 3. For instance, while the base VGG19 models are severely vulnerable, our randomized feature defense achieves 22.2% in robust accuracy after 10000 query, also significantly better than the randomized input defense (17.8% robust accuracy). On the transformer-based DeiT, our randomized feature defense has 69.1% robust accuracy under Square attack, while the robust accuracy of the randomized input defense is 2% lower. For the NES attack, the randomized-feature VGG19 shows the best robustness. In summary, randomized feature defense consistently achieves high robustness on most models except ResNet50 where the robustness is similar to randomized input defense.

Table 1: Defense Performance in ImageNet. The clean-data accuracy of the robust models is allowed to drop either $\approx 1\%$ or $\approx 2\%$.

| Model | Method | Acc | Square | | NES | | SignHunt | |
|---|---|---|---|---|---|---|---|---|
| | | | 1000 | 10000 | 1000 | 10000 | 1000 | 10000 |
| ResNet50 | Base | 80.37 | 3.5 | 0.2 | 36.2 | 4.3 | 6.6 | 0.4 |
| | Input | 79.18 ($\approx 1\%$) | 40.3 | 39.5 | 63.8 | 23.9 | 47.6 | 45.4 |
| | | 78.46 ($\approx 2\%$) | 41.1 | 39.8 | **69.4** | **41.5** | 49.3 | 47.2 |
| | Feature | 79.70 ($\approx 1\%$) | 37.0 | 36.0 | 56.7 | 16.8 | 46.3 | 43.4 |
| | | 78.43 ($\approx 2\%$) | **42.0** | **41.5** | 65.6 | 40.6 | **51.3** | **49.3** |
| VGG19 | Base | 74.21 | 0.1 | 0.0 | 19.6 | 0.0 | 0.4 | 0.0 |
| | Input | 73.24 ($\approx 1\%$) | 7.7 | 6.9 | 32.1 | 1.5 | 18.3 | 17.0 |
| | | 71.43 ($\approx 2\%$) | 18.7 | 17.8 | 47.4 | 11.5 | 28.3 | 27.1 |
| | Feature | 72.66 ($\approx 1\%$) | 22.4 | 21.6 | 50.1 | 18.5 | 34.6 | **32.9** |
| | | 71.21 ($\approx 2\%$) | **23.3** | **22.2** | **55.1** | **28.4** | **36.5** | 32.8 |
| DeiT | Base | 82.00 | 6.4 | 0.0 | 46.7 | 0.8 | 22.3 | 0.0 |
| | Input | 80.10 ($\approx 1\%$) | 67.7 | 67.2 | **75.8** | 65.9 | 64.4 | 63.6 |
| | | 79.60 ($\approx 2\%$) | 66.6 | 66.0 | 75.7 | **67.1** | 64.9 | 64.3 |
| | Feature | 80.80 ($\approx 1\%$) | **69.7** | **69.1** | 75.0 | 59.1 | **66.4** | 64.1 |
| | | 79.76 ($\approx 2\%$) | 69.3 | 69.0 | 75.1 | 65.3 | 66 | **64.3** |
| ViT | Base | 79.15 | 5.7 | 0.0 | 45.7 | 7.3 | 5.1 | 0.0 |
| | Input | 78.28 ($\approx 1\%$) | 58.8 | 58.1 | 70.8 | 51.4 | 53.1 | 52.2 |
| | | 77.09 ($\approx 2\%$) | 61.3 | 60.9 | 70.6 | **59.2** | 53.7 | 52.7 |
| | Feature | 78.20 ($\approx 1\%$) | 60.6 | 60.2 | 69.1 | 47.5 | 54.0 | 52.9 |
| | | 77.18 ($\approx 2\%$) | **63.7** | **62.9** | **72.2** | 58.1 | **57.0** | **55.3** |

Table 2: Defense Performance in CIFAR10. The clean-data accuracy of the robust models is allowed to drop either $\approx 2\%$ or $\approx 4\%$.

| Model | Method | Acc | Square | | NES | | SignHunt | |
|---|---|---|---|---|---|---|---|---|
| | | | 1000 | 10000 | 1000 | 10000 | 1000 | 10000 |
| ResNet50 | Base | 97.66 | 0.8 | 0.1 | 71.7 | 21.7 | 3.7 | 0.2 |
| | Input | 95.98 ($\approx 2\%$) | 50.5 | 48.8 | 93.1 | 85.4 | 26.8 | 26 |
| | | 93.42 ($\approx 4\%$) | 56.4 | **54.8** | 90.0 | 85.0 | 31.1 | 29.8 |
| | Feature | 95.95 ($\approx 2\%$) | 54.9 | 52.8 | **93.2** | **86.2** | 32.5 | 30.6 |
| | | 93.48 ($\approx 4\%$) | **56.7** | 53.4 | 89.9 | 83.9 | **37.1** | **35.7** |
| VGG19 | Base | 96.28 | 0.6 | 0.1 | 68.8 | 16.6 | 3.2 | 0.3 |
| | Input | 94.92 ($\approx 2\%$) | 30.6 | 27.1 | 89.5 | 58.0 | 22.7 | 21.8 |
| | | 93.52 ($\approx 4\%$) | 42.2 | 39.8 | 90.3 | 68.4 | 27.5 | 26.8 |
| | Feature | 94.93 ($\approx 2\%$) | 61.0 | 58.4 | **92.2** | 77.9 | 43.2 | 42.4 |
| | | 93.58 ($\approx 4\%$) | **64.2** | **62.8** | 91.2 | **80.1** | **49.2** | **46.9** |
| DeiT | Base | 98.40 | 3.2 | 0.0 | 81.9 | 34.2 | 7.9 | 0.2 |
| | Input | 96.59 ($\approx 2\%$) | 66.9 | 67.6 | **95.2** | **90.0** | 40.2 | 39.2 |
| | | 94.81 ($\approx 4\%$) | **70.6** | **68.8** | 92.6 | 87.7 | 40.3 | 38.5 |
| | Feature | 96.29 ($\approx 2\%$) | 69.1 | 67.9 | 94.1 | 88.3 | **45.7** | **43.4** |
| | | 94.91 ($\approx 4\%$) | 68.9 | 66.1 | 93.5 | 87.6 | 43.6 | 40.4 |
| ViT | Base | 97.86 | 5.1 | 0.0 | 84.8 | 43.6 | 6.1 | 0.0 |
| | Input | 95.80 ($\approx 2\%$) | 63.0 | 61.2 | 93.5 | **87.0** | 34.8 | 33.3 |
| | | 93.40 ($\approx 4\%$) | 62.6 | 61.1 | 89.7 | 85.5 | 33.4 | 32.2 |
| | Feature | 95.96 ($\approx 2\%$) | 63.9 | 62.7 | **93.7** | 85.6 | 42.5 | 40.7 |
| | | 93.39 ($\approx 4\%$) | **66.2** | **65.6** | 92.9 | 85.3 | **44.8** | **43.8** |

We also observe similar robustness results on CIFAR10 experiments with ResNet50, VGG19, DeiT, and ViT for 3 attacks. As we can observe in Table 2, randomized feature and input defenses are effective against score-based attacks. Similar to ImageNet, randomized feature defense achieves significantly better robustness than randomized input defense in most experiments. For Square attacks on ResNet50 and DeiT, while the best robustness is achieved by randomized input defense, randomized feature defense is more robust when the defender sacrifices 2% clean-data accuracy.

Table 3: Robustness (higher means more robust) under different values of $\mu$. **Small** $\nu$ corresponds to selected $\nu$ where clean accuracy is allowed to drop by 2%, and **Large** $\nu$ corresponds to clean accuracy drop of 4%.

| Attack | $\mu$ | VGG | | | | ViT | | | |
|---|---|---|---|---|---|---|---|---|---|
| | | Small $\nu$ | | Large $\nu$ | | Small $\nu$ | | Large $\nu$ | |
| | | Input | Feature | Input | Feature | Input | Feature | Input | Feature |
| Square | 0.05 | 30.6 | 61.0 | 42.2 | 64.2 | 63.0 | 63.9 | 62.6 | 66.2 |
| | 0.1 | 47.4 | 65.8 | 54.6 | 65.5 | 69.3 | 70.2 | 68.8 | 69.6 |
| | 0.2 | 32.1 | 59.7 | 43.9 | 64.0 | 56.1 | 58.0 | 56.8 | 58.6 |
| | 0.3 | 27.0 | 54.9 | 38.1 | 59.7 | 47.1 | 51.9 | 47.7 | 50.4 |
| NES | 0.001 | 93.4 | 93.9 | 90.1 | 91.4 | 93.7 | 94.8 | 90.3 | 93.5 |
| | 0.01 | 89.5 | 92.2 | 90.3 | 91.2 | 93.5 | 93.7 | 89.7 | 92.9 |
| | 0.1 | 88.0 | 90.0 | 86.7 | 89.6 | 87.9 | 91.4 | 86.7 | 90.6 |
| | 0.2 | 93.6 | 93.0 | 92.6 | 91.4 | 91.0 | 93.8 | 87.6 | 92.0 |
| SignHunt | 0.01 | 91.6 | 91.0 | 91.3 | 88.0 | 89.1 | 90.9 | 85.4 | 91.3 |
| | 0.05 | 22.7 | 43.2 | 27.5 | 49.2 | 34.8 | 42.5 | 33.4 | 44.8 |
| | 0.075 | 5.6 | 19.7 | 8.1 | 25.6 | 13.6 | 22.5 | 13.7 | 24.3 |
| | 0.1 | 1.2 | 7.9 | 2.4 | 12.1 | 5.5 | 11.3 | 5.2 | 12.7 |

**Dynamic Analysis of Robustness.** As the adversary increases the magnitude of perturbation, the attack becomes more effective since the misleading probability decreases as shown in Theorem 1. The adversary can vary the square size for Square attack, the exploration step for NES, and the budget for SignHunt (since SignHunt sets the finite-difference probe to the perturbation bound).

Table 3 reports the robustness of the models under stronger attacks from these adversaries for different values of $\nu$. We can observe that increasing the strength of the attack leads to lower robustness among all the defenses. However, at the selected defense noise scales corresponding to the same clean accuracy drop, our defense is still more robust than randomized input defense; this improved robustness again can be explained by the analysis in Section 3.3 and 3.5. A larger attack perturbation may also cause the approximation in the attack to be less accurate, which leads to a drop in the attack's effectiveness; for example, the robust-

Table 4: Robustness with adversarial training.

| | Square | NES | SignHunt |
|---|---|---|---|
| AT | 32.5 | 67.6 | 31.7 |
| Ours | 37.6 | 44.1 | 41.7 |
| Ours+AT | 77.8 | 80.6 | 67.0 |

Table 5: Robustness against decision-based attacks (CIFAR10)

| Model | Method | Acc | RayS | SignFlip |
|---|---|---|---|---|
| ResNet50 | Base | 97.66 | 0.1 | 20.5 |
| | AAA | 97.70 | 0.1 | 20.4 |
| | Input | 93.52 | 12.0 | **85.5** |
| | Feature | 92.10 | **14.4** | 82.5 |
| VGG19 | Base | 96.28 | 0.0 | 6.4 |
| | AAA | 96.30 | 0.1 | 5.7 |
| | Input | 93.42 | 8.1 | **86.0** |
| | Feature | 93.48 | **15.4** | 76.5 |

Table 6: Robustness in CIFAR10 at each layer (fixed $\nu$).

| Model | Layer | Square | NES | SignHunt | GradNorm |
|---|---|---|---|---|---|
| VGG | 1 | 56.7 | 87.8 | 21.5 | 1.324 |
| | 4 | 52.5 | 84.2 | 18.7 | 0.842 |
| | 12 | 63.0 | 89.7 | 29.4 | 2.514 |
| | 15 | 50.6 | 87.7 | 37.4 | 1.710 |
| ViT | 1 | 77.3 | 94.8 | 26.1 | 0.615 |
| | 4 | 75.3 | 94.4 | 28.0 | 0.462 |
| | 8 | 65.8 | 91.6 | 26.9 | 0.324 |
| | 11 | 48.3 | 86.5 | 23.1 | 0.214 |

Table 7: Defenses against adaptive attacks on CIFAR10

| Attacks | Methods | VGG19 | | | | | | ResNet50 | | | | | |
|---|---|---|---|---|---|---|---|---|---|---|---|---|---|
| | | Acc | $M = 1$ | $M = 5$ | | $M = 10$ | | Acc | $M = 1$ | $M = 5$ | | $M = 10$ | |
| | | | QC=1000 | QC=1000 | QC=5000 | QC=1000 | QC=10000 | | QC=1000 | QC=1000 | QC=5000 | QC=1000 | QC=10000 |
| Square | Input | 94.92 | 30.6 | 24.2 | 10.5 | 30.2 | 3.2 | 95.32 | 52.9 | 42.0 | 34.8 | 35.0 | 13.3 |
| | Feature | 94.93 | 61.0 | 53.0 | 45.5 | 46.7 | 23.1 | 95.21 | 54.5 | 45.1 | 40.4 | 37.3 | 21.1 |
| NES | Input | 94.92 | 89.5 | 93.4 | 82.1 | 94.4 | 78.8 | 95.32 | 92.4 | 94.0 | 91.3 | 93.9 | 90.7 |
| | Feature | 94.93 | 92.2 | 94.8 | 88.4 | 94.5 | 86.0 | 95.21 | 91.8 | 93.8 | 90.8 | 94.0 | 90.4 |
| SignHunt | Input | 94.92 | 22.7 | 15.9 | 10.4 | 23.3 | 7.6 | 95.32 | 29.9 | 17.6 | 13.5 | 21.1 | 9.4 |
| | Feature | 94.93 | 43.2 | 27.1 | 23.0 | 31.7 | 17.0 | 95.21 | 35.1 | 17.3 | 16.4 | 21.5 | 11.3 |

ness increases from $89.6\%$ to $91.4\%$ when the NES's perturbation magnitude increases in VGG19 experiments (similar observations in ViT).

**Combined with Adversarial Training (AT).** We evaluate the combination of our defense and AT on CIFAR10/ResNet20 model against under score-based attacks with 1000 queries and observe significantly improved robustness, as shown in Table 4.

### 4.3 PERFORMANCE AGAINST DECISION-BASED ATTACKS

Table 5 reports the performance of VGG19 and ResNet50 against 2 decision-based attacks on CIFAR10. Besides randomized feature and input defenses, we also include AAA defense, which optimizes the perturbation that does not change the prediction. While AAA is optimized for score-based attacks directly and thus is successful in fooling these attacks (as seen in Table 3 in Supplementary), the results show that AAA is not effective in defending against decision-based attacks, while randomized feature and input defenses improve the robustness. An interesting observation is that RayS attack is more effective than score-based attacks although it only uses hard labels, even when there are defenses.

### 4.4 RELATIONSHIP BETWEEN THE GRADIENT NORM AND THE ROBUSTNESS TO SCORE-BASED ATTACKS

In Table 6, we provide the corresponding accuracy under attack on CIFAR10 with 1000 queries (for when a single layer is randomized with a fixed value of $\nu$) and the mean of the gradient norm at that layer. As we can observe, as the gradient norm increases (also as we originally observed in Figure 1), the robustness also increases, thus verifying our theoretical results.

### 4.5 PERFORMANCE AGAINST ADAPTIVE ATTACKS

We conduct experiments with adaptive attacks that apply Expectation Over Transformation (EOT) Athalye et al. (2018) in which the attacker queries a sample $M$ times and averages the outputs to cancel the randomness. Table 7 show the robust accuracy of VGG19 and ResNet50 on CIFAR10 against EOT attack with $M = 5$ and $M = 10$. Note that with EOT, the number of updates in the attack is $M$ times less than that of a normal attack with the same query budget. For this reason, we report the results for adaptive attacks with both 1000 queries and $M \times 1000$ queries. We can observe that EOT can mitigate the effect of randomized defenses even with the same number of queries; however, feature defense still yields better performance.

## 5 CONCLUSION AND FUTURE WORK

In this work, we study the effectiveness of random feature defense against query-based attacks, including score-based and decision-based attacks. We provide an analysis that connects the robustness to the variance of noise and the local behavior of the model. Our empirical results show that random defense helps improve the performance of the model under query-based attacks with a trade-off in clean accuracy. Future works will be directed toward the analysis covering black-box attacks that transfer adversarial samples from the surrogate model to the target model.

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

APPENDIX

This Appendix provides additional details, analysis, and experimental results to support the main paper. We begin by discussing the limitations in Section A and societal impacts in Section B. Then, we provide the proof for Theorem 1 in the main paper in Section C. Next, the detailed experimental setup is provided in Section D, which is followed by additional robustness experiments to demonstrate the effectiveness of the proposed defense. Finally, we provide additional visualization of the robust behavior of the models with randomized features in Section F.

## A   LIMITATIONS

As discussed, we focus on studying the effectiveness of the randomized feature defense for DNNs against black-box attacks, including score-based and decision-based attacks. We do not include black-box attacks that utilize the transferability from surrogate to target models since our threat model does not make any assumptions about the network architecture and the training dataset.

Our proposed defense adds another layer of protection to DNNs against adversarial attacks, and it would be interesting to study the adversarial robustness of the combination of our randomized feature defense and existing defense strategies, including those that have been developed for transfer-based attacks Pang et al. (2019); Tramèr et al. (2018); Yang et al. (2020). We leave these to future works.

Similar to the related works on randomized defenses, our empirical evaluation has limitation for it is designed to assess the relative effectiveness of randomized models. From the adversary perspective, the attack should only need to make the system fail once, during the querying process. Correspondingly, it means that the attack can query the model repeatedly, and by chance, the defense fails at some point; however, this failure, equivalently meaning the attack is successful, is not due to the perturbed input being an adversarial example but rather comes from the added randomness of the defense. Here, there are 2 scenarios: (1) if the input (or the perturbed query) is far from the decision boundary, randomization is much less likely to shift the input to the other side of the boundary, making this chance very low. On the other hand, (2) for input close to the decision boundary, this "repeated" attack will be more effective, unfortunately; one potential solution is to preemptively stop this attack if the system recognizes the same input is repeatedly forwarded to the model. We leave this to future work and urge practitioners to research this inherent problem of randomized models.

In practice, an attack has a cost, and if the attack cost is higher than the potential gain, the attacker will more likely stop; thus a defense should increase this cost as much as possible. A randomized approach like ours increases the attack cost by confusing their optimization trajectory. Note that as explained earlier, if the perturbed query is in scenario (1) (or the randomized radius is small enough), our randomization would likely not make the attack successful due to chance; on the other hand, the defense would make it more costly for the attack to push this query to scenario (2).

## B   SOCIETAL IMPACTS

Deep neural networks (DNN) rapidly transform our daily lives in various domains and applications. Unfortunately, most well-trained DNNs are vulnerable to adversarial attacks, which decreases confidence in their deployment. Among the existing adversarial attacks, query-based attacks pose a severe threat to users since these attacks are effective and only require access to the model's feedback; the attackers do not know the trained parameters or model architectures.

Our work tackles this defense challenge against query-based attacks by proposing a lightweight adversarial defense for existing DNNs. We conduct a detailed theoretical analysis of our defense and show its superior performance compared to other randomized defenses in extensive empirical experiments across a wide range of DNN architectures, query-based attacks, and benchmark datasets. Most importantly, our method can be directly integrated into any existing off-the-shelf DNN. In summary, the proposed randomized feature defense can boost the adversarial robustness of existing DNNs against most query-based attacks, further improving the users's confidence when using them in practice.

## C  PROOF OF SECTION 3.2

**Theorem 1.** *Assuming the proposed random vector $u$ is sampled from a Gaussian $\mathcal{N}(0, \mu I)$, the model is decomposed into $f = g \circ h$, and the defense adds a random noise $\delta \sim \mathcal{N}(0, \nu I)$ to the output of $h$. At input $x$, the probability that the attacker chooses an opposite action positively correlates with*

$$\arctan\left(-\left(\frac{2\nu}{\mu} \frac{\|\nabla_{h(x)}(\mathcal{L} \circ g)\|_2^2}{\|\nabla_x(\mathcal{L} \circ f)\|_2^2}\right)^{-0.5}\right).$$

*Proof.* For score-based attacks, the attacker finds the direction by computing $\mathcal{L}(f(x+u)) - \mathcal{L}(f(x))$. When applying randomized defense, the direction instead relies on $\mathcal{L}(f_{\text{rand}}(x + u)) - \mathcal{L}(f_{\text{rand}}(x))$. As discussed in Section 3, the probability that the attacker chooses an opposite action is

$$\mathbb{P}\left[\frac{\mathcal{L}(f_{\text{rand}}(x + u)) - \mathcal{L}(f_{\text{rand}}(x))}{\mathcal{L}(f(x + u)) - \mathcal{L}(f(x))} < 0\right]. \tag{15}$$

If the defense adds a random noise $\delta \sim \mathcal{N}(0, \nu I)$ to the output of layer $h$ of the model $f = g \circ h$, we have $\mathcal{L}(f_{\text{rand}}(x)) = \mathcal{L}(g(h(x) + \delta))$. Since $\delta$ and $u$ are small, we can linearly approximate the objection function

$$\mathcal{L}(f(x + u)) \approx \mathcal{L}(f(x)) + \nabla_x(\mathcal{L} \circ f)^\mathsf{T} u, \tag{16}$$

$$\mathcal{L}(g(h(x) + \delta)) \approx \mathcal{L}(f(x)) + \nabla_{h(x)}(\mathcal{L} \circ g)^\mathsf{T} \delta, \tag{17}$$

$$\mathcal{L}(g(h(x + u) + \delta)) \approx \mathcal{L}(g(h(x) + J_h(x)u + \delta)) \tag{18}$$

$$\approx \mathcal{L}(f(x)) + \nabla_{h(x)}(\mathcal{L} \circ g)^\mathsf{T} J_h(x)u + \nabla_{h(x)}(\mathcal{L} \circ g)^\mathsf{T} \delta \tag{19}$$

$$= \mathcal{L}(f(x)) + \nabla_x(\mathcal{L} \circ f)^\mathsf{T} u + \nabla_{h(x)}(\mathcal{L} \circ g)^\mathsf{T} \delta, \tag{20}$$

where $\nabla_{h(x)}(\mathcal{L} \circ g)$ is the gradient of $\mathcal{L} \circ g$ evaluated at $h(x)$, $\nabla_x(\mathcal{L} \circ f)$ is the gradient of $\mathcal{L} \circ f$ at $x$, $J_h(x)$ is the Jacobian matrix of $h$ at $x$.

At a different application of $f_{\text{rand}}$, the randomized model samples a new noise vector. Let $\delta_1, \delta_2$ be the sampled noises when querying $\mathcal{L}(f_{\text{rand}}(x + u))$ and $\mathcal{L}(f_{\text{rand}}(x))$, the ratio can be approximated by

$$\frac{\mathcal{L}(f_{\text{rand}}(x + u)) - \mathcal{L}(f_{\text{rand}}(x))}{\mathcal{L}(f(x + u)) - \mathcal{L}(f(x))} \approx \frac{\nabla_x(\mathcal{L} \circ f)^\mathsf{T} u + \nabla_{h(x)}(\mathcal{L} \circ g)^\mathsf{T} \delta_1 - \nabla_{h(x)}(\mathcal{L} \circ g)^\mathsf{T} \delta_2}{\nabla_x(\mathcal{L} \circ f)^\mathsf{T} u} \tag{21}$$

$$= 1 + \frac{\nabla_{h(x)}(\mathcal{L} \circ g)^\mathsf{T}(\delta_1 - \delta_2)}{\nabla_x(\mathcal{L} \circ f)^\mathsf{T} u}. \tag{22}$$

Since $\delta_1$ and $\delta_2$ are independent, we have $\delta_1 - \delta_2 \sim \mathcal{N}(0, 2\nu I)$, thus

$$\nabla_{h(x)}(\mathcal{L} \circ g)^\mathsf{T}(\delta_1 - \delta_2) \sim \mathcal{N}(0, 2\nu\|\nabla_{h(x)}(\mathcal{L} \circ g)\|_2^2), \tag{23}$$

$$\nabla_x(\mathcal{L} \circ f)^\mathsf{T} u \sim \mathcal{N}(0, \mu\|\nabla_x(\mathcal{L} \circ f)\|_2^2). \tag{24}$$

The noises $\delta_1, \delta_2$ added by the defense and the noise $u$ added by the attacker are independent, therefore the ratio of two independent normal variables $\frac{\nabla_{h(x)}(\mathcal{L} \circ g)^\mathsf{T}(\delta_1 - \delta_2)}{\nabla_x(\mathcal{L} \circ f)^\mathsf{T} u}$ follows Cauchy distribution with location 0 and scale $\sqrt{\frac{2\nu}{\mu}} \frac{\|\nabla_{h(x)}(\mathcal{L} \circ g)\|_2}{\|\nabla_x(\mathcal{L} \circ f)\|_2}$. In this case, the probability that the attacker is fooled can be approximated by

$$\mathbb{P}\left[\frac{\mathcal{L}(f_{\text{rand}}(x + u)) - \mathcal{L}(f_{\text{rand}}(x))}{\mathcal{L}(f(x + u)) - \mathcal{L}(f(x))} < 0\right] \approx \mathbb{P}\left[\frac{\nabla_{h(x)}(\mathcal{L} \circ g)^\mathsf{T}(\delta_1 - \delta_2)}{\nabla_x(\mathcal{L} \circ f)^\mathsf{T} u} < -1\right] \tag{25}$$

$$= \frac{1}{\pi}\arctan\left(-\left(\frac{2\nu}{\mu} \frac{\|\nabla_{h(x)}(\mathcal{L} \circ g)\|_2^2}{\|\nabla_x(\mathcal{L} \circ f)\|_2^2}\right)^{-0.5}\right) + \frac{1}{2}. \tag{26}$$

$\square$

## C.1 Analysis of the Approximation Error

Due to the high nonlinearity of neural networks, it is difficult to have a complete analysis of the convexity of the boundary. However, if the model is $k_1-$Lipschitz, the change in the prediction when a noise $u$ is added is bounded by $k_1$ and $u$.

$$||f(x + u) - f(x)||_2^2 \le k_1 ||u||_2^2.$$

If $k_1$ is not too large, since $||u||_2^2$ is very small, in expectation the prediction of the randomized models stays still. Furthermore, we empirically validate whether the majority of the perturbed inputs belong to the original class, as shown in the experiment below.

For each input, we sample 20 pairs of perturbation with opposite directions. It's similar to antithetic sampling employed in Ilyas et al. (2018) for a better estimation of the effect of the injected noise. Intuitively, if the majority of the perturbed inputs are still in the original class, the average of the predictions is the same as the output of the original model. We also consider **the extreme case** (or worst case) where an input is marked as misclassified if *any of those* 40 noise vectors misleads the model. The table below shows the original accuracy, the average accuracy, and the extreme accuracy on VGG19/CIFAR10. As can be observed in Table 8, the decrease in accuracy of the expected case is trivial (0.37%); and even in the extreme case, this decrease is still small (2.3%) even in the extreme case. This supports our assumption.

Table 8: The accuracy of randomized defense

| Acc | Expected Acc | Extreme Acc |
| --- | --- | --- |
| 96.28 | 95.91 | 93.95 |

The perturbation added by the defense induces a negligible effect on the prediction of the model; since $\mu$ is constrained by the adversarial constraint (i.e., within the $L_p$ ball), it is typically smaller than $\nu$, thus inducing even small impact than $\nu$ on the prediction. Furthermore, if the gradient of the loss function $L(x)$ with respect to the input is $k_2-$Lipschitz continuous, we can bound the error of the approximation by $k_2$ and the norm of the noise $u$

$$L(x + u) \le L(x) + \nabla_x L \cdot u + \frac{1}{2} k_2 ||u||_2^2.$$

If $k$ and the noise are small, the error is also small. This assumption is also used in Qin et al. (2021) for their analysis. We also provided the error of the first-order approximation in Section F.1. The histogram of the error implies that the added noise is small enough for a close approximation, thus our analysis is valid.

# D Experimental Setup

## D.1 Dataset

In this work, we conduct experiments on two widely used datasets in adversarial attacks, CIFAR10 and ImageNet. We randomly selected 1000 images per dataset such that the test sets cover all classes, each of which has equal size.

- **CIFAR10**[1] consists of $60,000$ images from 10 different classes where the training set has $50,000$ images and the test set has $10,000$ images.
- **ImageNet (ILSVRC) 2012**[2] is a large-scale dataset that consists of 1000 classes. The training set includes $1,281,167$ images, the validation set includes $50,000$ images, and the test set has $100,000$ images.

For all experiments, we resize the images to $224 \times 224$ resolution.

---

[1] https://www.cs.toronto.edu/~kriz/cifar.html
[2] https://www.image-net.org/download.php

## D.2 Models

As discussed in the main text, we consider 4 models that have various architectures, including ResNet50 He et al. (2016), VGG19 Simonyan & Zisserman (2015), ViT base Dosovitskiy et al. (2021), DeiT base Touvron et al. (2021). We use the pretrained weights from `timm` package[3] for ImageNet, and finetune ResNet50, VGG19, ViT base, DeiT base for CIFAR10.

## D.3 Implementation of the Black-box Attacks

We perform experiments on 3 score-based black box attacks (Square attack Andriushchenko et al. (2020), NES Ilyas et al. (2018), and Signhunter Al-Dujaili & O'Reilly (2020)) and 2 decision-based attacks (RayS Chen & Gu (2020) and SignFlip Chen et al. (2020b)). For $\ell^\infty$ attacks, we find adversarial samples within the $\ell^\infty$ ball of radius 0.05, for $\ell^2$ attacks we set the radius to 5. The detailed hyperparameters of each attack are as follows:

- **Square attack**: The initial probability of pixel change is 0.05 for $\ell^\infty$ attack and 0.1 for $\ell^2$ attack.

- **NES**: We estimate the gradient by finite difference with 60 samples for $\ell^\infty$ attack and 30 for $\ell^2$ attack. The step size of finite difference is 0.01 and 0.005, and the learning rate is set to 0.005 and 1 for $\ell^\infty$ and $\ell^2$ attack, respectively.

## D.4 Evaluation

According to Section 3, a sample is considered as adversarial if it can fool the model in the majority of its application. Since the randomized model has stochasticity, for any datapoint there is a chance that the prediction is flipped at some application. Therefore, an attacker can stop the attack before finding the true adversarial sample. To alleviate this issue, when deciding whether to stop the attack, the query is repeated multiple times, and the attack is considered to be successful if the prediction is consistently flipped in most of the runs. The experiment applies 9 query runs for verification, and these extra runs are not included in the total number of queries.

# E  Additional experiments

## E.1 Performance against $\ell^2$ Attacks

We provide the results of randomized feature defense against $\ell^2$ attacks on CIFAR10 in Table 9. As we can observe, $\ell^2$ attacks are quite successful in fooling the model; however, randomized feature defense improves the robustness of the models to these attacks.

Table 9: The robustness against $\ell^2$ attacks on CIFAR10.

| Model | Method | Acc | Square | | NES | | SignHunter | |
|---|---|---|---|---|---|---|---|---|
| | | | 1000 | 10000 | 1000 | 10000 | 1000 | 10000 |
| VGG19 | Base | 96.28 | 13.5 | 3.0 | 72.9 | 50.7 | 44.2 | 11.2 |
| | Feature | 93.58 | 85.3 | 82.8 | 91.4 | 87.9 | 87.1 | 85.7 |
| ViT | Base | 97.86 | 48.5 | 28.6 | 87.8 | 73.5 | 60.7 | 30.3 |
| | Feature | 93.38 | 88.2 | 88.1 | 92.7 | 90.1 | 86.4 | 85.7 |

## E.2 Performance against Decision-based Attacks

Table 10 shows the performance of the model on ImageNet under decision-based attacks. Similar to CIFAR10, randomized feature defense is effective against decision-based attacks while AAA Chen

---

[3] `https://github.com/huggingface/pytorch-image-models`

et al. (2022) defense is not helpful in this case, since decision-based attacks only rely on the label that the model returns and AAA defense keeps the output label the same. We also provide the results for RayS on ViT/CIFAR10 in Table 11.

Table 10: The robustness against decision-based attacks on ImageNet.

| Model | Method | Acc | RayS | SignFlip |
|-------|--------|-----|------|----------|
| VGG19 | Base | 74.21 | 0.1 | 1.0 |
| | AAA | 74.24 | 0.4 | 0.8 |
| | Input | 71.43 | 6.6 | 53.3 |
| | Feature | 71.21 | 10.0 | 46.9 |
| ViT | Base | 79.15 | 1.7 | 9.7 |
| | AAA | 79.14 | 2.2 | 9.7 |
| | Input | 77.09 | 41.3 | 70.3 |
| | Feature | 77.18 | 41.0 | 70.1 |

**AAA's Performance against score-based attacks.** We also provide the evaluation of AAA under score-based attacks in Table 12. Since AAA is optimized for score-based attacks directly, it is successful in fooling the attack. However, under a general setting where the defender does not know the type of attack is currently performed (a more realistic scenario), AAA failed miserably as shown above, while our defense performs well regardless of the attack.

### E.3 Performance against White-box Attacks

As mentioned in the main paper, similar to previous works Byun et al. (2021); Qin et al. (2021); Chen et al. (2022), our threat model focuses on defending against black-box, query-based attacks, as it is a more realistic scenario in practice. Nevertheless, in this section, we provide an additional study about the performance of our defense against white-box attacks, those that require access to the model's architecture and its parameters. We evaluate the performance of our method against C&W Carlini & Wagner (2017) and PGD Madry et al. (2018) with $\ell_\infty$ constraint $\epsilon = 0.03$. As observed in Table 13, the proposed defense can boost the robustness against these white-box attacks while having a negligible degradation in clean accuracy. We conjecture that adding stochasticity to the model can transform it into a smoothed classifier and therefore reduce the adversarial effect.

### E.4 Robustness characteristics of layers

As discussed in Section 3.3 in the main paper, the gradient norm varies during the sequence of queries of an attack on an input. Figure 1 suggests that, in deeper layers, the ratio of the gradient norm increases during the attack, which is related to the model's robustness as seen in Theorem 1; thus the model becomes more resilient to black-box attacks. Here, we additionally provide the performance evaluation on CIFAR10 with 1000 attack queries when each layer is perturbed alone (with $\nu$ such that the clean accuracy drops within a similar threshold), as well as the mean rate of change in the gradient norm's ratios during the sequence of queries. Table 14 implies that deeper layers induce higher change and lead to better robustness, which confirms our analysis.

| Method | RayS |
|--------|------|
| Base | 2.0 |
| Input | 16.0 |
| Feature | 17.5 |

Table 11: The Accuracy under RayS attack of ViT on CIFAR10.

Table 12: Robustness against score-based attacks of AAA (CIFAR10).

| Attack | VGG19 | | | ViT | | |
|---|---|---|---|---|---|---|
| | Input | AAA | Feature | Input | AAA | Feature |
| Square | 18.7 | 27.1 | 23.3 | 61.3 | 66.6 | 63.7 |
| NES | 47.4 | 58.6 | 55.1 | 70.6 | 73.7 | 72.2 |

Table 13: Accuracy under white-box attacks of randomized feature defense on CIFAR10

| Model | Method | Acc | C&W | PGD |
|---|---|---|---|---|
| VGG19 | Base | 96.28 | 7.35 | 3.71 |
| | Feature | 94.93 | 34.93 | 6.06 |
| ViT | Base | 97.86 | 39.23 | 1.65 |
| | Feature | 95.96 | 55.11 | 26.01 |

### E.5 Performance When The Attack is Lucky

Our empirical experiments focus on evaluating whether the attack can truly find adversarial examples. For non-randomized models, when an attack arrives at the decision that $x$ is an "adversarial" example, $x$ unquestionably is on the other side of the decision boundary. However, for a randomized model, $x$ could be still on the correct side of the decision boundary, but the added randomization shifts it to the other side of the decision boundary; thus, in principle, $x$ is still not an adversarial example, and if we use 1 single application for evaluation, an attack could be lucky or a randomized defense could be unlucky. Consequently, for a fair evaluation of the effectiveness of a defense, we forward $x$ multiple times and decide that it is an adversarial example if the majority of the results say so, as seen in our paper.

Nevertheless, we understand that in the case where we ignore fair evaluation, the attack is allowed to be lucky and just has to fool the defense once. Therefore, we also provide the experiments when forwarding $x$ only once in Table 15. As we can observe, our defense is still effective against the attacks.

## F Behavior of Models with Randomized Features

### F.1 Approximation Error of Our Analysis

In our theoretical analysis, we apply first-order approximation to study the behavior of the model. To show that the error is negligible, we calculate the difference between the loss when the input is shifted by a vector $u$ and the value computed by first-order approximation. Figure 3 illustrates the

Table 14: Robustness (CIFAR10) at each layer with $\nu$ corresponding to $\approx 4\%$ clean accuracy drop.

| Model | Layer | Square | NES | SignHunt | Change of Ratio |
|---|---|---|---|---|---|
| VGG | 1 | 41.0 | 90.7 | 28.7 | 0.959 |
| | 8 | 68.9 | 90.7 | 46.4 | 1.273 |
| | 12 | 63.3 | 89.4 | 46.1 | 1.364 |
| | 15 | 55.9 | 87.4 | 41.1 | 1.318 |
| ViT | 1 | 67.5 | 89.7 | 17.6 | 1.089 |
| | 4 | 69.0 | 90.2 | 24.0 | 1.692 |
| | 8 | 69.5 | 89.8 | 37.2 | 1.751 |
| | 11 | 78.0 | 92.7 | 43.9 | 1.728 |

Table 15: The results of Square attack on ViT/ImageNet with one forward each iteration and budget query of $N$.

| Method | Acc | N=500 | N=1000 | N=5000 | N=10000 |
|--------|-----|-------|--------|--------|---------|
| Base | 79.15 | 13.4 | 10.3 | 0.2 | 0.0 |
| Input | 78.28 | 48.0 | 46.6 | 44.8 | 44.0 |
| Feature | 78.20 | 48.2 | 47.0 | 45.7 | 44.8 |

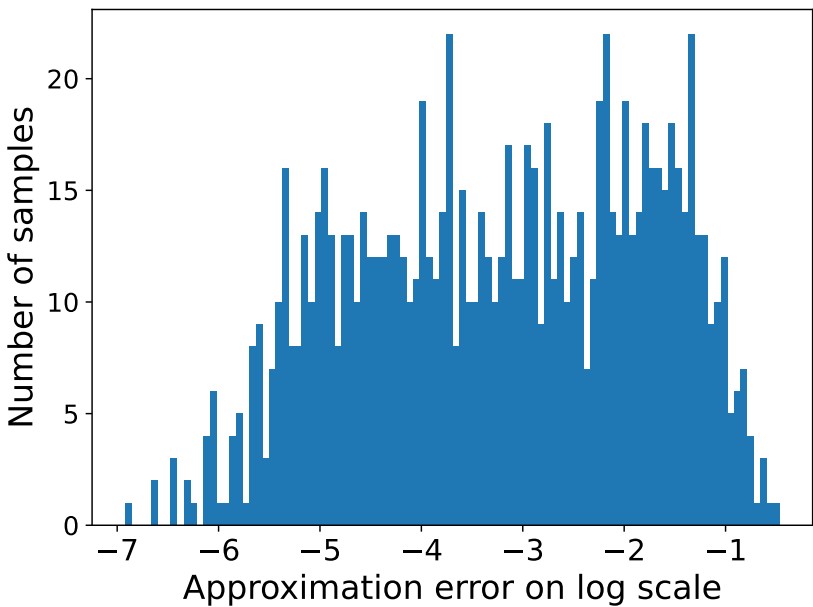

Figure 3: The error of first-order approximation of ViT on ImageNet at $\log_{10}$ scale.

histogram of the error of ViT on ImageNet at $\log_{10}$ scale, showing that our analysis is based on a good approximation.

## F.2 THE RATIO OF THE NORM OF THE GRADIENT

We report the ratio $\frac{2\nu}{\mu} \frac{\|\nabla_{h(x)}(\mathcal{L}\circ g)\|_2^2}{\|\nabla_x(\mathcal{L}\circ f)\|_2^2}$ before and after perturbed at hidden layers of VGG19 and ViT on ImageNet/CIFAR10 in Figure 6 and 7. The results show that, on both datasets, when the perturbed sample moves close to adversarial samples, the probability that randomized feature defense can fool the attacker increases while the probability of randomized input defense does not change. This explains the effectiveness of randomized feature defense against score-based attacks. Figure 7 shows this ratio on ViT. On ImageNet, the robustness of the defense still increases during the attack; however, on CIFAR10, such behaviors between the original and perturbed samples are not significantly different.

We also include the ratio on ResNet50 and DeiT on CIFAR10 in Figure 4 and Figure 5. As can be observed, the ratios on ResNet50 and DeiT do not increase much during the attack, leading to small improvements.

## F.3 THE MAGNITUDE OF THE ROBUSTNESS AT INPUT AND HIDDEN LAYERS

Figure 8 and 9 show the robustness of randomized feature and randomized input defenses by multiplying the norm of the gradient with the relative magnitude of the defense noise. As we can observe, the robustness when injecting noise to the hidden layers is generally higher than when injecting noise in the input. Such robustness behaviors are more visible in the deeper layers.

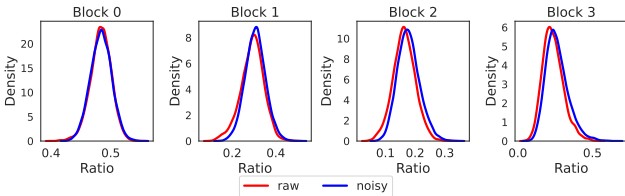

Figure 4: The ratio of the norm of the gradient of $\mathcal{L}$ at hidden layers and at input of ResNet50 on CIFAR10 before and after perturbed

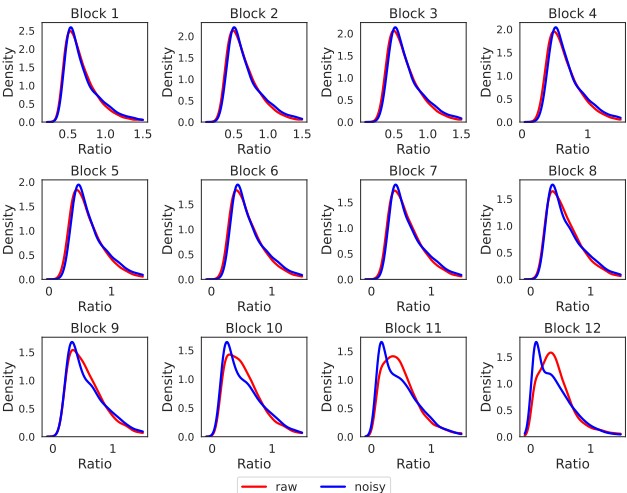

Figure 5: The ratio of the norm of the gradient of $\mathcal{L}$ at hidden layers and at input of DeiT on CIFAR10 before and after perturbed

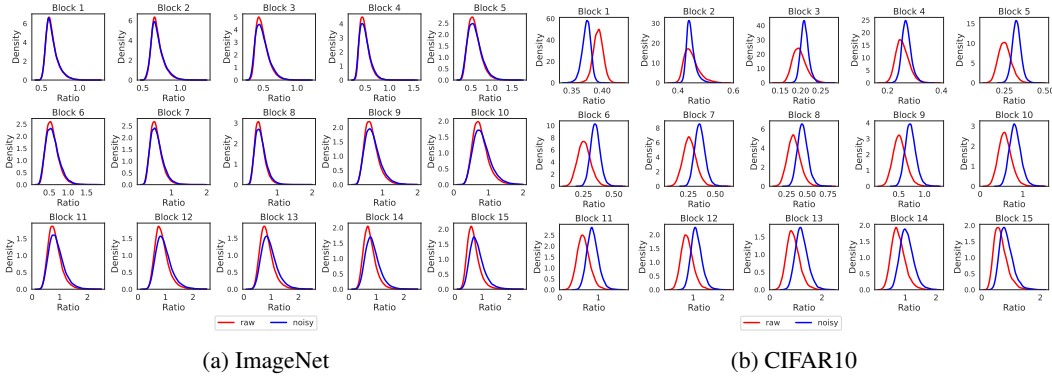

(a) ImageNet                    (b) CIFAR10

Figure 6: The ratio of the norm of the gradient of $\mathcal{L}$ at hidden layers and at input of VGG19 on ImageNet/CIFAR10 before and after perturbed

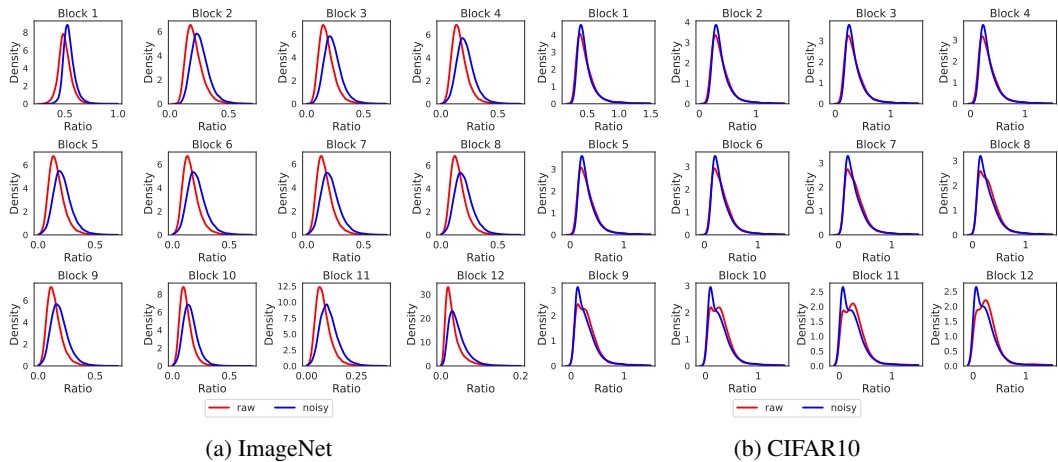

(a) ImageNet                                   (b) CIFAR10

Figure 7: The ratio of the norm of the gradient of $\mathcal{L}$ at hidden layers and at input of ViT on ImageNet/CIFAR10 before and after perturbed

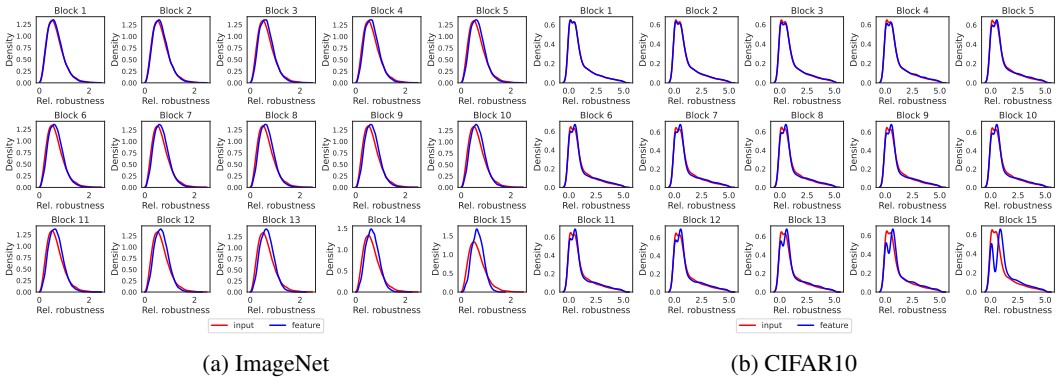

(a) ImageNet                                   (b) CIFAR10

Figure 8: Distributions of the magnitude of the robustness to query-based attacks computed at input and hidden layers of VGG19 on ImageNet/CIFAR10

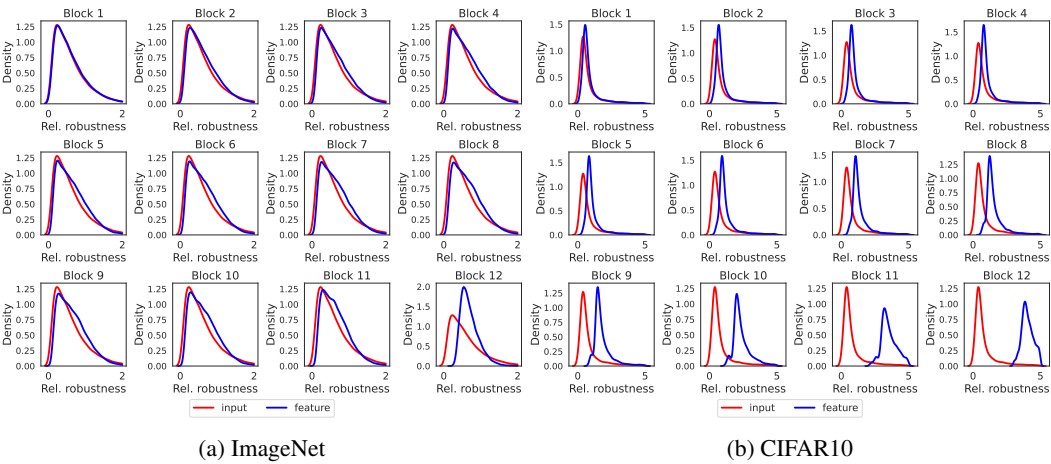

(a) ImageNet                                   (b) CIFAR10

Figure 9: Distributions of the magnitude of the robustness to query-based attacks computed at input and selected hidden layers of ViT on ImageNet/CIFAR10

