# OpenReview forum: "Understanding the Robustness of Randomized Feature Defense Against Query-Based Adversarial Attacks"
_ICLR.cc/2024/Conference — ICLR 2024 poster_

### Official Review · Reviewer_nGVZ · 2023-10-25

**Soundness:** 1 poor
**Presentation:** 1 poor
**Contribution:** 2 fair
**Rating:** 3
**Confidence:** 4

**Summary:**

This paper studies the defense method against query-based black-box attacks by injecting the noise into the middle layers of models. By theoretically analyzing the impact of both the adversarial perturbation and the noise injection on the prediction of the model, this paper tries to understand the impact of the proposed defense on robustness. Compared to the previous defense works that also inject noise into the model, the novelty of this paper is somehow in the noise injection to the feature space, i.e., the middle layer's outputs. Experimental results generally show the robustness improvement of injecting the noise to the feature rather than the input.

**Strengths:**

1. The proposed method that injects the noise into the features as the defense against the query-based black-box attack is novel and is empirically shown effective.

2. The adaptive attack is well-considered, which makes the evaluation more comprehensive.

**Weaknesses:**

1. The organization of this paper can be improved. Assumption 1, Figure 1 and 2 are not referred to in the main text. It is confusing on the purpose of presenting the assumption 1 and Figure 1 and 2.

2. The assumption and the theorem are incorrect. Even when the noise is small, the expectation of the randomized model is not necessarily consistent with the original model on the same data point, one simple counter-example is that when the input x is at the decision boundary, a small perturbation can change the prediction, so small noise may change the prediction. Theorem 1 is based on incorrect derivation, Eq. (23) and (24) may be incorrect as the gradient $\nabla_{h(x)}(L \cdot g)$ is anisotropic so the multiplication with Gaussian noise should not be an i.i.d. Gaussian noise. In addition, the assumption of the proof is the value of v and $\mu$ are small, so the approximation holds, but in the experiments, the value of $v$ is not present, and the value of $\mu$ is as large as 0.3, which is not negligible.

3. The correctness of Theorem 1 is not fully evaluated. The observation based on Theorem 1 is that the ratio $v/\mu$ is a factor of the robustness, if we fix the input x, then it is the only factor that affects the robustness. In Table 3, it is observed that the robustness is not strictly correlated to the ratio, this is reasonable since the inputs are changing during the multi-step perturbation. The correctness of the influence of the ratio can be verified by trying one-step perturbation so that the input x is kept the same, which is missing in this paper.

4. The evaluation of the decision-based attack is insufficient and the results are not good. It seems the proposed method only works on RayS, and the results on DeiT and ViT are not presented.

**Questions:**

1. Please verify if the Eq. (23) and the Eq. (24) are correct.

2. Please verify that assumption 1 is correct and that the theorems and experiments are strictly following this assumption.

3. I am curious about the impact of the ratio on the robustness when the gradients are fixed. Can you present the experimental results if possible?

---

> ### Author Response · Authors · 2023-11-20
> **Thank you for the valuable comments!**
>
> Please see our responses to your comments below:
>
> **Q1: Assumption 1, Figure 1 and 2 are not referred to in the main text. It is confusing on the purpose of presenting the assumption 1 and Figure 1 and 2.**
>
> A: Thank you for the observation. We have made incorrect references for Figures 1 and 2, when including the Supplementary File within the Main paper. Please see our clarifications below:
> - ‘Figure 3’ in Section 3.3 should be ‘Figure 1’,
> - ‘Figure 5’ in Section 3.5 should be ‘Figure 2’,
> - The discussion for Assumption 1 is immediately above its declaration. It allows us to ensure that small noise diffuses but does not shift the prediction, or adversarial samples of the original model are adversarial samples of the randomized model.
> We have updated this discussion in the revised submission.
>
> **Q2: The assumption and the theorem are incorrect. Even when the noise is small, the expectation of the randomized model is not necessarily consistent with the original model on the same data point, one simple counter-example is that when the input x is at the decision boundary, a small perturbation can change the prediction, so small noise may change the prediction.**
>
> A: As discussed in Section 3.2,  given Assumption 1, adversarial samples of the original model are adversarial samples of the randomized model, on expectation. In other words, at the decision boundary, while a specific sample of the small Gaussian noise may change the prediction, but expectedly, it should be the same prediction as that of the original model (across several samples of the small Gaussian noise). This assumption is typically made in other relevant randomization works, including the evaluated RND defense.
>
> **Q3: Theorem 1 is based on incorrect derivation, Eq. (23) and (24) may be incorrect as the gradient $\nabla_{h(x)}(L \cdot g)$ is anisotropic so the multiplication with Gaussian noise should not be an i.i.d. Gaussian noise.**
>
> A: Thank you for the comment. Please see our response below, confirming that it is **still indeed an i.i.d. Gaussian noise**.
>
> For a random vector $x$ following $\mathcal{N}(\mu, \Sigma)$, we have $Mx\sim\mathcal{N}(M\mu, M\Sigma M^T)$ which is still Gaussian noise. In Equation 23, since $\delta_1 - \delta_2\sim\mathcal{N}(0, 2\nu I)$ and $\nabla_{h(x)}(L \cdot g)2\nu I \nabla_{h(x)}(L \cdot g)^T=2\nu ||\nabla_{h(x)}(L \cdot g)||^2_2$, we have $\nabla_{h(x)}(L \cdot g)(\delta_1 - \delta_2)\sim\mathcal{N}(0, 2\nu ||\nabla_{h(x)}(L \cdot g)||^2_2)$.  Since the noise added by the attack and the defense at each application are independent, $\nabla_{h(x)}(L \cdot g)(\delta_1 - \delta_2)$ follows i.i.d Gaussian noise.
>
> A similar derivation is applied to Equation 24.
>
> **Q4: In addition, the assumption of the proof is the value of \nu and \mu are small, so the approximation holds, but in the experiments, the value of \nu is not present, and the value of \mu is as large as 0.3, which is not negligible.**
>
> A: In our experiments, as indicated (e.g., Tables 1 and 2), the selected $\nu$  corresponds to a small drop in accuracy (1-2%). These $\nu$ values are still very small, with magnitude less than 0.016. Our experiments show that the user can control the degree of clean accuracy loss for their application when selecting $\nu$, and typically, the user will not use a large $\nu$ value that will cause significant degradation in clean accuracy.
>
> On the other hand, $\mu$ is the step size that the attacker uses to search for the correct perturbed direction. For Square attack, the maximum square size is 0.3, however, the added noise is still bounded by the $\ell_{\infty}$ ball of radius 0.05. For NES $\mu$ is the step size of finite difference method, which should be small so the attacker can obtain a good approximation for the gradient; when $\mu$ is too large, this approximation becomes unreliable. In the experiments, we run the attacks with small exploration step size, in particular for Square attack 1 and 0.005 for $\ell_{2}$ and $\ell_{\infty}$ NES attacks, respectively. Note that, these values are still small, to ensure gradient approximation is reliable. These exact values have also been used in previous works, including RND and SND.

---

> > ### Author Response · Authors · 2023-11-20
> > **Continued rebuttal comments!**
> >
> > **Q5: The evaluation of the decision-based attack is insufficient and the results are not good. It seems the proposed method only works on RayS, and the results on DeiT and ViT are not presented.**
> >
> > A: Similar to RND, our method indeed doesn’t work well only on RayS/CIFAR10 with VGG19 and ResNet50 networks, as already acknowledged in Section 4.3. However, both RND and ours are significantly more effective in all other decision-based experiments, including the additional experiments with DeiT and ViT networks (results provided below, as suggested by the reviewer). Please note that AAA, another popular defense for score-based attacks that ours and RND also are effective against, is not effective at all for both RayS and SignFlip.
> >
> > In general, we believe that bulletproof protection of a model against adversarial attacks is an extremely difficult endeavor; to the best of our knowledge, no single defense is able to be effective against all different types of attacks in all tested scenarios. Nevertheless, as demonstrated in the paper, our method and RND both can provide protection against a broad range of attacks in the majority of scenarios, including score-based attacks, decision-base attacks, EoT attacks, white-box attacks, etc…
> >
> > We also believe that the best protection is to combine various types of defenses, consequently combining the strengths of each defense. We demonstrate this potential direction in Section 4.2 (Table 4), where combining our method and Adversarial Training significantly boosts the performance in all experiments from 30-40% accuracy under attack (of each method separately) to 70-80% accuracy under attack.
> >
> > *Decision-based attacks on DeiT and ViT on ImageNet with 10k iterations*
> >
> > | |         | RayS| SignFlip|
> > |----------|---------|------|----------|
> > | ViT      | Base    |  1.7 |      9.7 |
> > |          | Input   |   41 |     70.3 |
> > |          | Feature | 40.3 |     70.2 |
> > | DeiT| Base    |  3.7 |      5.9 |
> > |          | Input   | 52.5 |     75.5 |
> > |          | Feature | 49.7 |     73.7 |
> >
> > *EOT decision-based attacks with 2k iterations*
> >
> > |  |         | RayS| SignFlip|
> > |--------------|---------|------|----------|
> > | ViT      | Base    |  7.5 | 39.5  |
> > |              | Input   | 39.3 | 70.9 |
> > |              | Feature | 41.3 | 72.3 |
> > | DeiT| Base    | 12.8 | 40.7  |
> > |              | Input   | 52.6 | 76.5  |
> > |              | Feature | 54.7 | 75.5  |
> >
> > **Q6: Please verify if the Eq. (23) and the Eq. (24) are correct.**
> >
> > A: Please see the answer to Question 3.
> >
> > **Q7: Please verify that assumption 1 is correct and that the theorems and experiments are strictly following this assumption.**
> >
> > A: Please see the answer for Questions 2 & 3.
> >
> > **Q8: I am curious about the impact of the ratio on the robustness when the gradients are fixed. Can you present the experimental results if possible?**
> >
> > A: As provided in our analysis, the robustness of our randomization depends on the attack radius $\mu$, the defense noise scale $\nu$, and the ratio of the gradients. While $\mu$ and $\nu$ are fixed after the attack launches the attack and the defender set up the defense, respectively, the gradients are input-dependent. At each query step of an attack on an input, the perturbed input changes, thus the corresponding gradients also change. This is a distinct characteristic of our defense, compared to RND, and we show that the deeper the randomized layer is, its corresponding ratio is higher, leading to a more robust model.

---

> > > ### Comment · Reviewer_nGVZ · 2023-11-23
> > >
> > > Thanks for the detailed response. The response addresses my concerns in Q1 and Q3, while the rest of my concerns are not addressed well.
> > >
> > > For Q2, it is possible that the majority of the noise-perturbed inputs lie in the original class. To validate this assumption, you may need to provide some theoretical analysis of the convexity of the decision boundary.
> > >
> > > For Q4, it is hard to justify how small of $\mu$ and $\nu$ is enough for the approximation. It would be helpful if the authors could provide some theoretical analysis to quantify the error w.r.t. the magnitude of $\mu$ and $\nu$, otherwise, the qualitative analysis is not convincing.
> > >
> > > For Q5, the authors claim that the proposed method does not work well only on RayS/CIFAR10 with VGG19 and ResNet50 networks, it seems like the main text and the rebuttal still do not present the ViT and DeiT's results on CIFAR10 as evidence.
> > >
> > > My main concern is still on the validation of the assumption since this paper focuses on providing some theoretical guarantee to the defense, which largely relies on the assumption. If the assumption is not strictly satisfied, even if other papers have relied on it, the main contribution would be largely affected. A valid theoretical analysis of the correctness of the assumption would raise my rating.

---

> > > > ### Author Response · Authors · 2023-11-23
> > > > **Thank you for the new comments. Please see our responses below.**
> > > >
> > > > **Q2.1: it is possible that the majority of the noise-perturbed inputs lie in the original class. To validate this assumption, you may need to provide some theoretical analysis of the convexity of the decision boundary.**
> > > >
> > > > Thank you for your suggestion. Due to the high nonlinearity of neural networks, it is difficult to have a complete analysis of the convexity of the boundary. However, if the model is $k_1-$Lipschitz, the change in the prediction when a noise $u$ is added is bounded by $k_1$ and $u$.
> > > > $$||f(x+u)-f(x)||_2^2\leq k_1||u||_2^2.$$
> > > > If $k_1$ is not too large, since $||u||^2_2$ is very small, in expectation the prediction of the randomized models stays still. Furthermore, we empirically validate whether the majority of the perturbed inputs belong to the original class, as shown in the provided experiment in this response.
> > > >
> > > > For each input, we sample $20$ pairs of perturbation with opposite direction. It's similar to antithetic sampling employed in Ilyas et al. (2018) for a better estimation on the effect of the injected noise. Intuitively, if the majority of the perturbed inputs are still in the original class, the average of the predictions is the same as the output of the original model. We also consider **the extreme case** (or worst case) where an input is marked as misclassified if *any of those $40$* noise vectors misleads the model. The table below shows the orginal accuracy, the average accuracy, and the extreme accuracy on VGG19/CIFAR10. As can be observed, the decrease in accuracy of the  expected case is trivial (0.37%); and even in the extreme case, this decrease is still small (2.3%) even in the extreme case. This supports our assumption.
> > > >
> > > > | Acc | Expected Acc | Extreme Acc|
> > > > |---| ---| ---|
> > > > | 96.28 | 95.91 | 93.95 |
> > > >
> > > > Finally, we want to thank the reviewer again for allowing us to confirm the validity of our assumption, empirically. We included this analysis in the revised submission (Section C.1).
> > > >
> > > > **Q4.1:  it is hard to justify how small of $\mu$ and $\nu$ is enough for the approximation. It would be helpful if the authors could provide some theoretical analysis to quantify the error w.r.t. the magnitude of $\mu$ and $\nu$, otherwise, the qualitative analysis is not convincing.**
> > > >
> > > > As has been addressed in Q2.1, the perturbation added by the the defense induces negligible effect to the prediction of the model; since $\mu$ is constrained by the adversarial constraint (i.e., within the $L_p$ ball), it is typically smaller than $\nu$, thus inducing even small impact than $\nu$ on the prediction. Furthermore, if the gradient of the loss function $L(x)$ wrt the input is $k_2-$Lipschitz continuous, we can bound the error of the approximation by $k_2$ and the norm of the noise $u$
> > > > $$L(x+u)\leq L(x) + \nabla_xL\cdot u+\frac{1}{2}k_2||u||^2_2.$$
> > > > If $k$ and the noise are small, the error is also small. This assumption is also used in Qin et al. (2021) for their analysis. We also provided the error of the first-order approximation in Section F.1/Supplementary of the revised submission. The histogram of the error implies that the added noise is small enough for a close approximation, thus our analysis is valid.
> > > >
> > > > **Q5.1:  the authors claim that the proposed method does not work well only on RayS/CIFAR10 with VGG19 and ResNet50 networks, it seems like the main text and the rebuttal still do not present the ViT and DeiT's results on CIFAR10 as evidence.**
> > > >
> > > > Thank you for the comment. Due to limited time left on the discussion period, we could only provide the results for ViT. Our defense still higher AuA than RND (Input) and the Base model.
> > > >
> > > > |      |         | RayS |
> > > > |------|---------|------|
> > > > | ViT  | Base    | 2.0  |
> > > > |      | Input   | 16.0 |
> > > > |      | Feature | 17.5 |
> > > >
> > > > We have added this experiment to the revised submission.
> > > >
> > > > ---
> > > >
> > > > We hope that our responses to these questions can address your concerns. If you have any additional questions, please let us know and we will address them promptly. Finally, thank you very much for giving us the comments that helped us improve our paper.

---

### Official Review · Reviewer_s19G · 2023-10-30

**Soundness:** 3 good
**Presentation:** 3 good
**Contribution:** 2 fair
**Rating:** 6
**Confidence:** 4

**Summary:**

This paper investigates a well-known defense against black-box adversarial attacks (both score-based and decision-based) which involves adding a random noise to the input. The paper argues that the robustness-accuracy trade-off of such defense can be improved by adding noise to the intermediate layer instead. Theoretical and empirical analyses are provided to support this claim.

---

## Comment After Rebuttal

Once again, thank you so much for acknowledging and addressing my concerns! I appreciate your efforts.

Based on the new results (counting one successful attack query as a successful attack), there seems to be a minimal improvement from adding noise at the feature vs at the input. However, the result does show that the defense is effective in this difficult practical setting (~40% of samples are still correctly classified after 10k queries), and this convinces me that there are applications where this type of defense can be successfully applied.

I would really appreciate it if the author(s) could include this type of results and discussion (along with other suggestions during this rebuttal period) in the next revision of the paper. After reading the other reviewers' comments, I have no further concerns and have decided to adjust my rating from 5 to 6.

**Strengths:**

### Quality

The experiments are thorough, and the metrics are well-designed. Many models, datasets, and attack algorithms are included in the experiments. I like that the other baseline defense like AAA is also included. I also appreciate the comprehensive background section.

The paper also takes into account the nuance of picking the appropriate noise variance; they nicely solve this issue using the notion of the robustness-accuracy trade-off and pick the variance $\nu$ that results in a small fixed drop in the clean accuracy.

**Weaknesses:**

### Disadvantages of randomized models

I understand that the paper focuses on randomized models, but in practice, randomized models can be unattractive for two reasons:

1. Its output is stochastic and unpredictable. For users or practitioners, what a randomized model entails is the fact that all predictions have some unaccounted chance of being wrong. One may argue that it is possible to average the outputs across multiple runs, but doing so would reduce the robustness and just converge back to the deterministic case (and with increased costs).
2. **Definition of a successful attack**. This has to do with how a successful attack is defined on page 4: “…adversarial attacks are successful if the obtained adversarial example can fool the randomized model in the *majority* of its applications on the example.” I argue that in security-sensitive applications (e.g., authentication, malware detection, etc.), it is enough for the model to be fooled *once*. The randomness enables the attacker to keep submitting the same adversarial examples until by chance, the model misclassifies it.

I believe that these practical disadvantages limit the significance of this line of work.

### First-order Taylor approximation

All the analysis in the paper uses the first-order Taylor approximation. First of all, this assumption should be stated more clearly. More importantly, I am not convinced that this approximation is good especially when the added noise or the perturbation is relatively large. Neural networks are generally highly nonlinear so I wonder if there is a way to justify this assumption better. An empirical evaluation of the approximation error would make all the analyses more convincing.

### Method for picking the intermediate layer

First, I wonder which intermediate layer is picked for all the results in Section 4. Do you pick the best one empirically or according to some metric? It will also be good to clearly propose a heuristic for picking the intermediate layer and measure if or how much the heuristic is inferior to the best possible choice.

### Loss-maximizing attacks

One of the big questions for me is whether the attack is related to the fact that most of the black-box attacks try to **minimize the perturbation magnitude**. Because of the nature of these attacks, all the attack iterations stay very close to the decision boundary, and hence, they perform particularly poorly against the noise addition defense. In other words, these attacks are never designed for a stochastic system in the first place so they will inevitably fail.

The authors have taken some steps to adapt the attacks for the randomized defense, mimicking obvious modifications that the real adversary might do (EoT and Appendix D.4). I really like these initiatives and also wonder if there are other obvious alternatives. One that comes to mind is to use attacks that **maximize loss given a fixed $\epsilon$ budget**. These attacks should not have to find a precise location near the decision boundary which should, in turn, make it less susceptible to the randomness.

This actually does NOT mean that the randomness is not beneficial. Suppose that the loss-maximizing attack operates by estimating gradients (via finite difference) and just doing a projected gradient descent. One way to conceptualize the effect of the added noise is a noisy gradient, i.e., turning gradient descent into *stochastic* gradient descent (SGD). SGD convergence rate is slowed down with larger noise variance so the adversary will have to either use more iterations or uses more queries per step to reduce the variance. Either way the attack becomes more costly. I suggest this as an alternative because it directly tests the benefits of the added noise without exploiting the fact that the distance-minimizing attacks assume deterministic target models.

### Additional figures

I have suggestions on additional figures that may help strengthen the paper.

1. **Scatter plot of the robustness vs the ratio in Theorem 1.** The main claim of the paper is that the quantity in Theorem 1 positively correlates with the failure rate of the attack (and so the robustness). There are some figures that show the distribution of this quantity, but the figure that will help empirically verify this message is to plot it against the robustness (perhaps average over the test samples). Then, a linear fit and/or an empirical correlation coefficient can also be shown. Personally, this plot more clearly confirms the theoretical result than the density plot (e.g., Figure 2, etc.) or Table 6. I also think that $\nu$ should not be fixed across layers/models and should be selected to according to the clean accuracy.
2. **Scatter plot of the clean accuracy vs the ratio in Eq. (14)**. Similar to the first suggest, I would like to see an empirical confirmation for both of these theoretical analysis.
3. **Robustness-accuracy trade-off plot**. This has been an important concept for evaluating any adversarial defense. I would like to see this trade-off with varying $\nu$ as well as varying intermediate layers. The full characterization of the trade-off should also help in choosing the best intermediate layer, instead of just considering a few fixed values of $\nu$.

### Originality

One other weakness of the paper is the originality/novelty of the method. The most important contribution of this paper is the analysis on the gradient norm (i.e., sensitivity of the model) of benign and perturbed samples. The proposal to add noise to the intermediate layer instead of the input in itself is relatively incremental. However, the theoretical analysis does seem particularly strong to me, even though it does build up a nice intuition of the scheme. This is a minor weakness to me personally, and I would like to see more empirical results, as suggested earlier, rather than additional theoretical analyses.

### Other minor issues

- Eq. (7): the RHS is missing $L(...,y)$.
- 2nd paragraph on page 7: I think it was slightly confusing the first time I read it. It was not immediately clear to me that this is about Eq. (14) and what “product of this value and …” refers to.

**Questions:**

1. The paper mentions that the attack success rate is measured by “majority.” There’s an issue that I have already mentioned above, but I would like to know how many trials are used to compute this majority for the reported robustness in the experiments. If some variance can be reported too, that would be great.
2. Section 3.3: from my understanding, the ratios plotted in Figure 3 involve both $\nu$ and $\nabla_{h(x)}(L \circ g)$. I wonder how $\nu$ is picked here. Is it picked like in Section 3.5 where the accuracy drop is fixed at some threshold (e.g., 1%, 2%)?
3. Section 4.1: it is mentioned that Qin et al. [2021] and Byun et al. [2021] are included in the comparison, but they never show up again. I wonder if the results are really missing or if these two schemes are basically the noise addition in the input.
4. Generally, I see a larger improvement for VGG and ViT vs ResNet-50 and DeiT. Is there any explanation or intuition the authors can provide to better understand this observation?

---

> ### Author Response · Authors · 2023-11-20
> **Thank you for the valuable comments!**
>
> Please see our responses to your comments below:
>
> **Q1: randomized output is stochastic and unpredictable... a randomized model entails all predictions have some unaccounted chance of being wrong. average the outputs across multiple runs, but doing so would reduce the robustness and just converge back to the deterministic case (and with increased costs).**
>
> A: Randomized models have been extensively studied in the literature as an effective way to defend against adversarial attacks (Liu et al. 2017; He et al. 2019; Cohen et al. 2019; Salman et al. 2019; Byun et al. 2021; Qin et al. 2021). Indeed, since randomization may cause unpredictable predictions, existing works require training and model ensembles, both of which are expensive, to ensure minimal impact of the noise (i.e., cause a decrease in clean accuracy), while achieving high robustness. In contrast, our defense, and also RND, allow the user to control how much random noise to add to ensure a minimal drop in clean accuracy (using a small test set). For example, our defense achieves high robustness in Tables 1 and 2 against several attacks while only causing 1-2% accuracy drops in performance. Please note that defense approaches such as adversarial training cause a significant drop in clean accuracy (e.g., 10% or more in ViT (Mo, Yichuan, et al)) to achieve good robustness.
>
> Mo, Yichuan, et al. "When adversarial training meets vision transformers: Recipes from training to architecture." NeurIPS 2022.
>
> **Q2: Definition of a successful attack. I argue that in security-sensitive applications (e.g., authentication, malware detection, etc.), it is enough for the model to be fooled once**
>
> A: Our empirical experiments focus on evaluating whether the attack can truly find adversarial examples. For non-randomized models, when an attack arrives at the decision that $x$ is an adversarial example, $x$ unquestionably is on the other side of the decision boundary. However, for a randomized model, $x$ could be still on the correct side of the decision boundary, but the added randomization shifts it to the other side of the decision boundary; thus, in principle, $x$ is still not an adversarial example, and if we use 1 single application for evaluation, an attack could be lucky or a randomized defense could be unlucky. Consequently, for a fair evaluation of the effectiveness of a defense, we forward $x$ multiple times and decide that it is an adversarial example if the majority of the results say so, as seen in our paper.
>
> Nevertheless, we understand that in the case where we ignore fair evaluation, the attack is allowed to be lucky and just has to fool the defense once, as given in the example in the comment. Therefore, we also provide the experiments when forwarding $x$ only once, as below. As we can observe, our defense is still effective against the attacks.
>
> *Robustness with a single trial, 10k queries, 1%-accuracy noise scale, ImageNet*
> |  |       | Square  | NES     |
> |--------------|-------|---------|---------|
> | ViT          | Base | 0 | 7.3 |
> |              | Input | 44.0| 47.0 |
> |              | Feature | 44.8| 42.5 |
>
> **Q3: All the analysis in the paper uses the first-order Taylor approximation. An empirical evaluation of the approximation error would make all the analyses more convincing.**
>
> A: Thank you for your suggestion. We include the histogram of the approximation error of ViT on ImageNet in Section F.1 (Supplementary). As we can observe, the error is relatively small.
>
> **Q4: First, I wonder which intermediate layer is picked for all the results in Section 4. empirically or according to some metric?**
>
> We pick the penultimate layers of the base models for randomization, in all experiments. We have shown with theoretical analysis in Figure 1 (main) and empirical analysis in Table 12 (supplement), that the randomizing at a deeper layer consistently achieves the best robustness; for example, randomizing the last layer in VGG or ViT achieves 8-10% improvement over the previous layer. We have already revised the paper (Section 4.1) to clarify this discussion accordingly.

---

> ### Author Response · Authors · 2023-11-20
> **Continued rebuttal comments!**
>
> **Q5: The authors have taken some steps to adapt the attacks for the randomized defense, mimicking obvious modifications that the real adversary might do (EoT and Appendix D.4). I really like these initiatives and also wonder if there are other obvious alternatives. One that comes to mind is to use attacks that maximize loss given a fixed $\eps$ budget. These attacks should not have to find a precise location near the decision boundary which should, in turn, make it less susceptible to randomness.**
>
> **This actually does NOT mean that the randomness is not beneficial. Suppose that the loss-maximizing attack operates by estimating gradients (via finite difference) and just doing a projected gradient descent. One way to conceptualize the effect of the added noise is a noisy gradient, i.e., turning gradient descent into *stochastic* gradient descent (SGD). SGD convergence rate is slowed down with larger noise variance so the adversary will have to either use more iterations or more queries per step to reduce the variance. Either way, the attack becomes more costly. I suggest this as an alternative because it directly tests the benefits of the added noise without exploiting the fact that the distance-minimizing attacks assume deterministic target models.**
>
> A: Thank you for your appreciation of our analysis and indeed a great observation regarding the fixed budget. In fact, NES, Square attack, and SignHunt, all of which are evaluated in our paper, work in the principle of maximizing loss given a budget. Specifically, NES approximates the gradient directly and applies projected gradient descent, while Square attack and SignHunt search for the adversarial sample at the boundary of the $\ell_p$ ball.
>
> **Q6: additional figures: Scatter plot of the robustness vs the ratio in Theorem 1. Scatter plot of the clean accuracy vs the ratio in Eq. (14). Robustness-accuracy trade-off plot**
>
> Thank you for your suggestions. In fact, the robustness averaged over the test samples at each layer, corresponding to Figures 1 and 2, are provided in Table 12 (Supplementary). We can observe an increase in robustness as the ratio becomes higher at deeper layers. We have updated these Figures, in the revised paper, with the performance results created in Table 12, for a better explanation.
>
> As discussed in Section 4, our defense allows the user to choose $\nu$ that results in a “controlled” decrease in clean accuracy, using a small test set. For example, we provided the results of varying $\nu$ in Tables 1 and 2 that correspond to 1%, 2%, 3% and 4%. drops in clean accuracy. In general, increasing $\nu$, or more randomness, will lead to a decrease in clean accuracy but also an increase in robustness, as observed in these tables. In addition, we provide the robustness (Accuracy under Attack) when varying $\nu$ (corresponding varying drops in clean accuracy) on CIFAR10/VGG19/Square Attack setting:
> AuA = [56.2, 58.4, 62.3, 62.8, 63.5]
> Acc = [95.33, 94.94, 94.55, 93.58, 92.09]
>
> **Q7: The most important contribution is the analysis on the gradient norm (i.e., sensitivity of the model) of benign and perturbed samples. The proposal to add noise to the intermediate layer instead of the input in itself is relatively incremental. However, the theoretical analysis does seem particularly strong to me, even though it does build up a nice intuition of the scheme. This is a minor weakness to me personally, and I would like to see more empirical results, as suggested earlier**
>
> A:  Thank you for appreciating our work. Here, we summarize our responses to the earlier questions:
> - Evaluation with a single application on the successful query (answered in Q2, added to the revised paper)
> - Error analysis of first-order approximation (answered in Q3, added to the revised paper)
> - Performance at different layers with the same drop in accuracy (answered in Q6, provided in Table 12 of original submission)
> - Robustness-Accuracy tradeoff (answered in Q6 with additional experiments)
> Different from existing randomized-feature defenses, which additionally require adversarial expensive training and ensemble, our work opens the possibility of randomizing the latent layers at **inference time** without these hassles (being lightweight and plug-and-play) and with a strong theoretical foundation. In addition, unlike randomizing the input, such as RND, our defense gives the user protection against non-continuous input, such as graph data; however, this is beyond the scope of our paper and deserves another independent study.

---

> > ### Author Response · Authors · 2023-11-20
> > **Continued rebuttal comments!**
> >
> > **Q8: The paper mentions that the attack success rate is measured by “majority.” There’s an issue that I have already mentioned above, but I would like to know how many trials are used to compute this majority for the reported robustness in the experiments. If some variance can be reported too, that would be great.**
> >
> > A: Taking computational resources and statistical significance into account and following the previous work, Liu et al 2017 (with ~9 ensemble models), we choose to query the model 9 times to determine whether an input is adversarial.
> >
> > **Q9: Section 3.3: from my understanding, the ratios plotted in Figure 3 involve both \nu and `\nabla_{h(x)}(L \circ g)`. I wonder how \nu is picked here. Is it picked like in Section 3.5 where the accuracy drop is fixed at some threshold (e.g., 1%, 2%)?**
> >
> > A: Yes, we compute the ratios $\mathcal{L}(f(x), y)/||\nabla_h (\mathcal{L}\circ g)||$ for all samples and take the value that is higher than $1\%$ of all the computed ratios. This value is similar to inducing a 1% decrease in accuracy.
> >
> > **Q10: Section 4.1: it is mentioned that Qin et al. [2021] and Byun et al. [2021] are included in the comparison, but they never show up again. I wonder if the results are really missing or if these two schemes are basically the noise addition in the input.**
> >
> > A: The input randomization approach in RND (Qin et al. 2021) and SND (Byun et al. 2021) are identical; Qin et al. [2021] provide more rigorous theoretical support for randomized input. For this reason, the performance evaluation of SND is exactly the same as in the report for RND. We have updated Section 2.3 of the paper to clarify this discussion.
> >
> > **Q11: Generally, I see a larger improvement for VGG and ViT vs ResNet-50 and DeiT. Is there any explanation or intuition the authors can provide to better understand this observation?**
> >
> > A: As shown in the analysis, the robustness depends on the ratio of the gradient and this value changes during the attack. While these ratios increase significantly on VGG and ViT, we plot the distribution of those on ResNet50 and DeiT in Section F2 of the revised submission and observe that these values do not increase much, explaining why the improvements are not as high as VGG/ViT. Studying why there exists such differences between the architectures beyond the scope of our paper and thus is left to future works.

---

> > > ### Comment · Reviewer_s19G · 2023-11-20
> > > **Reviewer s19G's response to the author rebuttal**
> > >
> > > I really appreciate the author’s clarification of my questions. My concerns/questions from Q3-7 are satisfactorily addressed. I have some responses and clarification for Q1 and Q2.
> > >
> > > **Practical motivation (Q1/2)**
> > >
> > > I am aware of the other prior works that also rely on randomness to combat query-based attacks, but I am not convinced that it is practically useful for the reasons I outlined in my review. It may, of course, be *scientifically* interesting, but I see this line of problem as *practically motivated*.
> > >
> > > I also did not mention anything about clean accuracy; I understand that to improve security, some trade-off must be made. I know that randomized defense usually has a much better clean accuracy than adversarial training. My first issue is about the unpredictability or the randomness itself which entails the security Implication in the second issue.
> > >
> > > Please allow me to rephrase the core issue I’m raising. The adversarial attack usually matters to security-sensitive applications (e.g., malware, self-driving cars) where *the system should not fail even once*. By allowing the system to be non-deterministic, I argue that we have increased that probability (e.g., the attacker just has to query the model repeatedly, and by chance, it will fail at some point!). On the other hand, if I use adversarial training, I may take a hit on clean accuracy, but at least, I can systematically study where my model fails and manage that risk rather than leaving things to chance.
> > >
> > > This issue is inherent to any randomized defense, and I don’t see a way to reconcile this. In other words, I have not found a way to practically justify this line of defense. That said, I’m more than happy to be swayed by your argument on this matter.
> > >
> > > **What I think is a good evaluation scheme**
> > >
> > > With the above reasoning, I would not call the evaluation by averaging over multiple queries “fair” at all. If anything, it is *unfair* because it’s more computationally costly and defeats the purpose of randomness. What’s considered “reasonable” to me is the attacker should be allowed to query $N$ times (including running the attack), and if *any* one of these $N$ queries leads to a successful attack, then the defender loses on this sample. This is what I expect practitioners or industry to do! Choosing an appropriate $N$ depends on applications, and it can be anywhere from 1 to ~10k. So I would report on multiple values of $N$ in that range.
> > >
> > > Note that what I propose here is different from letting the attacker complete the attack first and then evaluate only the final adversarial example in one query. It is unclear to me what is the exact procedure the authors used for the new result so I’d appreciate a clarification.

---

> > > > ### Author Response · Authors · 2023-11-22
> > > > **Responses to additional questions**
> > > >
> > > > **Q1.1 Practical motivation (Q1/2)**
> > > >
> > > > **Please allow me to rephrase the core issue I’m raising. The adversarial attack usually matters to security-sensitive applications (e.g., malware, self-driving cars) where the system should not fail even once. By allowing the system to be non-deterministic, I argue that we have increased that probability (e.g., the attacker just has to query the model repeatedly, and by chance, it will fail at some point!). On the other hand, if I use adversarial training, I may take a hit on clean accuracy, but at least, I can systematically study where my model fails and manage that risk rather than leaving things to chance.**
> > > >
> > > > **This issue is inherent to any randomized defense, and I don’t see a way to reconcile this. In other words, I have not found a way to practically justify this line of defense. That said, I’m more than happy to be swayed by your argument on this matter.**
> > > >
> > > > We really appreciate this comment for the perspective on randomized models. Indeed, evaluating randomized models has challenges due to their non-deterministic nature. In fact, we discussed this issue in Section 3.5, albeit from the accuracy perspective. To make this discussion more rigorous, we extend Section A to include the discussion on this thread; our hope is to ensure that the readers do not get a wrong message when looking at the robustness results provided in the paper.
> > > >
> > > > More specifically, we discuss that, similar to the related works on randomized defenses, our empirical evaluation is designed to assess the *relative effectiveness* of randomized models. Nevertheless, from the adversary perspective, the attack should only need to make the system fail once, during the querying process. Correspondingly, it means that the attack can query the model repeatedly, and by chance, the defense fails at some point; however, this failure, equivalently meaning the attack is successful, is not due to the perturbed input being an adversarial example, but rather comes from the added randomness of the defense. We also discuss 2 scenarios: (1) if the input (or the perturbed query) is far from the decision boundary, randomization is much less likely to shift the input to the other side of the boundary, making this chance very low. On the other hand, (2) for input close to the decision boundary, this "repeated" attack will be more effective, unfortunately; one potential solution is to preemptively stop this attack if the system recognizes the same input is repeatedly forwarded to the model. We leave this to future work and urge practitioners to research this inherent problem of randomized models.
> > > >
> > > > We think that it's very challenging to secure the model if we don't consider the cost of the attack. In practice, if the attack cost is higher than the potential gain, the attacker will more likely stop; thus a defense should increase this cost as much as possible. A randomized approach like ours increases the attack cost by confusing their optimization trajectory. Note that as explained earlier, if the perturbed query is in scenario (1) (or the randomized radius is small enough), our randomization would likely not make the attack successful due to chance; on the other hand, the defense would make it more costly for the attack to push this query to scenario (2).
> > > >
> > > > Finally, both directions (adversarial training and randomization) have pros and cons and we believe their usage depends on the application scenario.

---

> > > > > ### Author Response · Authors · 2023-11-22
> > > > > **Responses to additional questions (Continued)**
> > > > >
> > > > > **Q2.1 What I think is a good evaluation scheme**
> > > > >
> > > > > **What’s considered “reasonable” to me is the attacker should be allowed to query  $N$ times (including running the attack), and if any one of these $N$ queries leads to a successful attack, then the defender loses on this sample. This is what I expect practitioners or industry to do! Choosing an appropriate $N$ depends on applications, and it can be anywhere from 1 to ~10k. So I would report on multiple values of in that range.**
> > > > >
> > > > > **It is unclear to me what is the exact procedure the authors used for the new result so I’d appreciate a clarification.**
> > > > >
> > > > > For the new results (in Q2), we let the attack query the model and within the budget, if the attack can find any adversarial perturbation, we declare that the attack is successful. For example, the attack generate the perturbations $x_1, x_2, ..., x_n$ of a sample $x$ with label $y$, if $f(x_i) \ne y$ for any $i$ and $n\le$ query budget, the attack is successful. We believe this is the same as suggested.
> > > > >
> > > > > The table below evaluates the randomized defense with multiple query budgets, using this procedure. Relatively, the robustness of our defense is still slightly better than that of RND (or Input). As we vary the query budget, the robustness decreases, as expected.
> > > > >
> > > > > |        |         | Acc   | N=500 | N=1000 | N=5000 | N=10000 |
> > > > > |--------|---------|-------|-------|--------|--------|---------|
> > > > > | Square | Base    | 79.15 | 13.4  | 10.3   | 0.2    | 0.0     |
> > > > > |        | Input   | 78.28 | 48.0  | 46.6   | 44.8   | 44.0    |
> > > > > |        | Feature | 78.20 | 48.2  | 47.0   | 45.7   | 44.8    |
> > > > >
> > > > > The randomness of the randomized model can lead to the following two "by chance" cases: (1) $f(x_i)!=y$ due to randomization although $x_i$ is not an adversarial example, which marks the attack as "successful". Another case is (2) $f(x_i)==y$ also due to randomization although $x_i$ is an adversarial example, which marks the attack as "unsuccessful". This mostly happens when $x_i$ is close enough to the decision boundary (w.r.t randomization radius), and therefore is more sensitive to randomization of a randomized defense. Our original evaluation procedure was designed to reduce the impact of these cases when comparing between different randomization methods.

---

### Official Review · Reviewer_NLrk · 2023-11-01

**Soundness:** 3 good
**Presentation:** 3 good
**Contribution:** 2 fair
**Rating:** 6
**Confidence:** 4

**Summary:**

This paper proposes to defend against black-box attacks by adding noise to intermediate features at test time. It is empirically validated effective against both score-based and decision-based attacks. The authors also provide theoretical insights on the proposed method.

**Strengths:**

1. The idea is straightforward, lightweight, and can be plugged into all existing defenses like adversarial training.
2. It is great to see the theoretical analysis for the defense method.
3. The paper is well-organized and easy to follow.
4. The authors do comprehensive experiments to study the effectiveness of the proposed method.

**Weaknesses:**

1. The motivation to inject feature noise is not clear compared to injecting input noise. "Unlike previous randomized defense approaches that solely rely on empirical evaluations to showcase effectiveness" is not correct, since RND also provides lots of theoretical analysis as the authors acknowledged in Sec. 2.3. The results are not significantly better than RND, but injecting feature noise requires a careful choice of the layer.

2. The idea of injecting noise into hidden features is not novel, seeing Parametric Noise Injection: Trainable Randomness to Improve Deep Neural Network Robustness against Adversarial Attack, CVPR 2019. Although this is for defending against white-box attacks, adopting it for black-box attacks does not seem a significant contribution.

3. Does the proposed method have an advantage against AAA in defending score-based attacks? AAA is not designed for decision-based attacks, where the authors use AAA for comparison.

**Questions:**

Response to rebuttal: The authors provide a strong rebuttal and a good revision of the paper. My Q1 and Q3 have been well addressed, making me raise my score to 6. Although the method differs from the CVPR 2019 paper in Q2, the novelty is weak, i.e., perturbing feature to defend has been explored for a long time.

---

> ### Author Response · Authors · 2023-11-20
> **Thank you for the valuable comments!**
>
> Please see our responses to your comments below:
>
> **Q1: The motivation to inject feature noise is not clear compared to injecting input noise. "Unlike previous randomized defense approaches that solely rely on empirical evaluations to showcase effectiveness" is not correct, since RND also provides lots of theoretical analysis as the authors acknowledged in Sec. 2.3. The results are not significantly better than RND, but injecting feature noise requires a careful choice of the layer.**
>
> A: Thank you for the suggestion. In the above sentence, we referred to the lack of rigorous theoretical analysis of the robustness against black-box, query-based attacks when randomizing the internal features of the model. Similar to our work on the feature space, RND indeed provides a theoretical analysis on the input space. We have already revised the paper (Section 1) to clarify this claim accordingly.
>
> As we can observe in Tables 1 and 2, our randomized feature defense consistently outperforms RND (in several cases, the improved robustness of our defense is almost 10-20% more than that of RND) for all evaluated attacks, except NES, where RND occasionally performs better than our method (in these cases, RND’s slightly better robustness of ~2-3%).
>
> Finally, regarding the comment that “our method requires careful choice of the layer”:
> * We have shown with theoretical analysis in Figure 1 (main) and empirical analysis in Table 12 (supplement), that the randomizing at a deeper layer consistently achieves the best robustness; for example, randomizing the last layer in VGG or ViT achieves 8-10% improvement over the previous layer.
> * In fact, all our experiments randomize only the penultimate layers of the models. Consequently, the complexity of using our method is equivalent to that of RND. We already provided clarification on this claim (Section 4.1) in the revised submission.
> * Please note that an advantage of our defense over RND is that randomizing an internal layer makes the method suitable for discrete data, such as graphs. However, this is beyond the scope of our paper and deserves an independent study.
>
> **Q2: 1. The idea of injecting noise into hidden features is not novel, seeing Parametric Noise Injection: Trainable Randomness to Improve Deep Neural Network Robustness against Adversarial Attack, CVPR 2019. Although this is for defending against white-box attacks, adopting it for black-box attacks does not seem a significant contribution.**
>
> A: Although injecting noise to the features is previously proposed (such as in the mentioned paper), these methods require additional training to tune the noise, which is computationally expensive, especially when dealing with large and complex datasets. In contrast, our defense, and also RND, are lightweight, plug-and-play and can be applied to any pretrained model without additional model training. Our defense is effective against black-box attacks because it is designed to fool the querying process of these attacks, as seen in our theoretical and empirical analysis.
>
> **Q3: Does the proposed method have an advantage against AAA in defending score-based attacks? AAA is not designed for decision-based attacks, where the authors use AAA for comparison.**
>
> A:  We already provided the experiments, comparing AAA to our defenses on score-based attacks in Section E.3 (Supplementary). The experiments show that that our defense improves the robustness and has comparable results to AAA on score-based attacks. However, for decision-based attacks, as provided in Section 4.3, our defense has superior performance. From a practical perspective, a defense should work well against many types of query-based black-box attacks, and these experiments show that our defense has more practical utility than AAA.

---

### Official Review · Reviewer_pa1M · 2023-11-01

**Soundness:** 4 excellent
**Presentation:** 3 good
**Contribution:** 4 excellent
**Rating:** 8
**Confidence:** 3

**Summary:**

This paper showed that adding noises to some parts of models could protect the models from query-based attacks. The authors derived proofs to show that their method (adding noises) theoretically provided robustness to the models. Besides, they experimented this method with several datasets (Imagenet and CIFAR10) and models' architectures (i.e., ResNet50, VGG19, DeiT and ViT).

**Strengths:**

- The paper has a strong theoretical proof to show that the method can effectively provide robustness.
- The experiments are strong because the authors used Imagenet and CIFAR10 to show that their method and generalize in small and large datasets. Also, they tried with several models' architectures.

**Weaknesses:**

- I understand that the paper focuses on black-box attacks, but in the experiment section, the authors may try evaluating models with white-box attacks as well.
- Please check the parentheses in equation (7).

**Questions:**

- In page 5, can you please give a reason for this sentence "We can observe that these ratios become higher when the data are perturbed toward the adversarial samples. In other words, the randomized model is more robust during the attack."?

---

> ### Author Response · Authors · 2023-11-20
> **Thank you for the valuable comments!**
>
> Please see our responses to your comments below:
>
> **Q1: I understand that the paper focuses on black-box attacks, but in the experiment section, the authors may try evaluating models with white-box attacks as well.**
>
> A: Our work focuses on studying the effect of randomized models on black-box attacks, which are more practical in MLaaS system. In these systems, the adversary rarely has access to the trained model or model’s architecture to perform white-box attacks. Although our analysis does not cover white-box attacks, in Section E.4 we still conduct experiments for white-box attacks on randomized feature defense. The experimental results on C&W and PGD show that our defense can also boost the model’s robustness against white-box attacks, which demonstrates the broad utility of our method beyond black-box attacks.
>
> **Q2: Please check the parentheses in equation (7).**
>
> A: Thank you for the comment. We have fixed the typo in the revised submission.
>
> **Q3: In page 5, can you please give a reason for this sentence "We can observe that these ratios become higher when the data are perturbed toward the adversarial samples. In other words, the randomized model is more robust during the attack."?**
>
> A: In our theoretical analysis, we show that the robustness to black-box attacks is controlled by three terms: the scale of noise vector added by the attack $\mu$ and the defense $\nu$, and the ratio of the norm of the gradient with respect to the feature where noise is added and the input $\frac{||\nabla_h(L\circ g)||}{||\nabla_h(L\circ g)||}$. While the scales of the noise vectors are fixed during an attack, the last term is dependent on the input and where in the model the defense is performed.  During the attack, the input is perturbed and thus changes the above ratio if noise is injected into hidden layers of the model. Figure 1 illustrates that the ratio increases, especially for deeper randomized layers, when the attack happens, implying that randomized feature defense has a higher chance than input defense to fool black-box attacks.

---

> > ### Comment · Reviewer_pa1M · 2023-11-22
> >
> > Thank you for your answers and clarification!

---

### Author Response · Authors · 2023-11-20
**Thanks the reviewers for the initial evaluation of our paper!**

We would like to express gratitude to the reviewers for their initial comments and questions. We have responded to each of the questions from the reviewers and updated our submission to incorporate your valuable feedback (our revisions are in red). During the discussion period, we are happy to answer any additional questions you may have. Again, thank you!

---

### Author Response · Authors · 2023-11-23
**Thank you again for helping improve our work!**

We propose a lightweight and plug-and-play randomized feature defense and provide both theoretical analysis of the randomization effect and comprehensive empirical analysis of its effectiveness across a wide range of attacks, including black-box query-based attacks (score-based, decision-based), adaptive attacks, and white-box attacks, under various evaluation setups. We also show that, empirically, our randomization adds a trivial effect on the predictions of the model.

During the rebuttal process, we have:
- Provided experimental details and the rationale to choose the layers to perform the defense.
- Provided our perspective why randomized defense are practical and various evaluation metrics/setups (e.g., single and multiple applications on the success query) to evaluate these defenses; we acknowledge that there are inherent challenges in evaluating randomized models.
- Demonstrated the robust-accuracy trade-off of our defense.
- Provided initial theoretical analysis and emprical evidences to justify the assumption that the randomized defense induces negligible effect to the prediction of the model, to additionally support our analysis in the original submission.
- Provided the error of first-order approximation, showing that our analysis is applicable to neural networks.
- Provided initial theoretical analysis to quantify the error of the approximation.
- Clarified that the proof of our theorem is correct.
- Provided additional experiments for decision-based attacks on ViT/DeiT for CIFAR10/ImageNet.

We have added these discussion in the revised submissions. We hope that our responses have addressed the concerns of the reviewers. Finally, we're happy to promptly answer additional questions.

---

### Meta-Review · Area_Chair_vLsT · 2023-12-12

**Metareview:**

This submission received mixed rating. The rebuttal was convincing for some reviewers and they engaged with the authors. Reviewer nGVZ rated at level 3 and is not fully convinced after the rebuttal period. However, since other reviewer are in favor of accepting the paper, AC believes it is a good idea to accept it. The main weaknesses include limited novelty compared to CVPR 2019 paper.

The main weaknesses include limited novelty compared to CVPR 2019 paper. Basically, the idea of perturbing features to defend has been explored in the prior work and has limited novelty. Also, the results are only marginally better than state-of-the-art methods. Hence, AC believes acceptance as a poster is the right decision for this submission.

**Justification For Why Not Higher Score:**

The main weaknesses include limited novelty compared to CVPR 2019 paper. Basically, the idea of perturbing features to defend has been explored in the prior work and has limited novelty. Also, the results are only marginally better than state-of-the-art methods. Hence, AC believes acceptance as a poster is the right decision for this submission.

**Justification For Why Not Lower Score:**

The submission has interesting idea and strong results, and most reviewers suggest accepting it.

---

### Decision · Program_Chairs · 2024-01-16

Accept (poster)